# The effect of batch size on contrastive self-supervised speech representation learning

## Abstract

Foundation models in speech are often trained using many GPUs, which implicitly leads to large effective batch sizes. In this paper we study the effect of batch size on pre-training with the contrastive method wav2vec 2.0, both in terms of statistics that can be monitored during pre-training, and in the effect on the performance of downstream fine-tuning tasks. By using batch sizes varying from 87.5 seconds to 80 minutes of speech we show that, for a fixed amount of iterations, larger batch sizes result in better pre-trained models, with an upper limit for effectiveness. We then show that the quality of the pre-trained model depends mainly on the amount of speech data seen during training, i.e., on the product of batch size and number of iterations. Our extensions can help researchers choose effective operating conditions when studying self-supervised learning in speech, and hint towards benchmarking self-supervision with a fixed amount of seen data. Code and model checkpoints are available at `https://github.com/anonymous/available-after-review`.

## 1 Introduction

Foundation models have become the norm in deep learning research. In the audio domain, popular models with open weights include wav2vec 2.0 (Baevski et al., 2020; Conneau et al., 2021), HuBERT (Hsu et al., 2021), and WavLM (Chen et al., 2022b). These transformer models all use a form of self-supervised learning (SSL) with the use of a pretext task to learn ("pre-train") speech representations. The models can then be fine-tuned on a myriad of downstream tasks (Yang et al., 2021), including speech recognition, speaker recognition, emotion recognition, and intent classification. However, self-supervised pre-training takes a tremendous amount of resources, exceeding high-end consumer grade hardware at the time of writing. First, due to the unlabeled nature of self-supervision, it is relatively cheap to increase the dataset size. Over a span of two years, we have seen public training datasets increase by two orders of magnitude[1], with wav2vec 2.0 using 1 k hours of audio (circa 100 GB) from Librispeech (Panayotov et al., 2015), to WavLM using 94 k hours (circa 10 TB) by combining Libri-light (Kahn et al., 2020), GigaSpeech (Chen et al., 2021) and VoxPopuli (Wang et al., 2021). Secondly, the seminal works mentioned above all report results of models trained with large batch sizes using data parallelism across many GPUs. For example, for models with 94 M parameters, WavLM and HuBERT use 32 GPUs, and wav2vec 2.0 uses 64 GPUs, with batch sizes of respectively 3 hours, 45 minutes, and 90 minutes of audio. The number of GPUs needed to work with these large batches in a timely manner, as well as the required disk space for the datasets, make it non-trivial to apply these algorithms.

While the effect of dataset size on performance is (at least partially) known (Baevski et al., 2020; Conneau et al., 2021), to our knowledge there are no studies on the scaling behaviour of SSL algorithms with respect to the batch size and number of training iterations. This can be of interest to researchers who do not have the resources to study these algorithms under large batch size conditions, or practitioners who need to make a trade-off between time, computational budget, and desired performance. Given a fixed model complexity, dataset size, and number of training iterations, how much is gained by increasing the batch size? How well

---

[1] Leaving aside Whisper (Radford et al., 2023) and Google USM (Zhang et al., 2023), with respectively 680 k hours and 12 M hours of private training datasets.

do these techniques work with fewer resources, and can the academic community do meaningful experiments without industrial-scale data centers? While we aim to answer these questions generally, for precisely the reason of available computational resources, we limit ourselves to studying the wav2vec 2.0 (Baevski et al., 2020) model extensively, leaving other SSL methods for future work. Concretely, we set out to address the following research questions:

RQ 1: How does the batch size affect the pre-training procedure of wav2vec 2.0?

RQ 2: How does the batch size during pre-training of wav2vec 2.0 affect downstream fine-tuning?

RQ 3: Can we compensate for a reduction of the batch size by increasing the amount of training iterations by the same factor?

Regarding all three RQs, and given the existing literature on speech SSL (Baevski et al., 2020; Hsu et al., 2021; Chen et al., 2022b), our hypothesis is that large batch sizes are essential for pre-training convergence and the model's ability to be properly fine-tuned to the downstream task. For RQ 1, we are interested in knowing whether a large batch size is a necessity for optimizing the objective. It would be valuable to know the smallest possible converging batch size, and how optimization behaves with this batch size compared to the canonical, large batch size. For RQ 2, we expect that a larger batch size will lead to better downstream task performance, but we are especially interested in how much the performance improves with each doubling of the batch size. What is the minimum batch size at which we see that fine-tuning is possible, and how does this depend on the amount of data available for fine-tuning? For RQ 3, we are interested in knowing whether training twice as long with half the batch size results in the same performance. There is evidence that contrastive methods benefit from large batch sizes (Chen et al., 2022a), although in wav2vec 2.0 the quantity of potential negatives samples does not increase with the batch size, only with the sequence length. Moreover, according to work on optimizer scaling laws (Goyal et al., 2018; Malladi et al., 2022), with a fixed number of epochs, learning rates can be adjusted accordingly with the batch size to obtain a very similar optimization trajectory. Therefore, we hypothesize that performance for wav2vec 2.0 is only a function of how much data is seen during self-supervision, and that with patience, people with fewer resources can also carry out pre-training.

To answer these questions, we pre-train wav2vec 2.0 with batch sizes ranging from 87.5 seconds to 80 minutes. We then fully fine-tune these models for speech recognition (updating most weights), with 10 minutes to 960 hours of labeled speech. To include other speech technology tasks, we also fine-tune following the SUPERB benchmark protocol, where small downstream models are trained on different categories of speech tasks. Here, the foundation model weights are frozen, and the (trainable) weighted sum of all layer outputs of the foundation model are used as input. Hereby, we make the following contributions:

1. We perform a comprehensive study of the effect of batch size and amount of training iterations for pre-training wav2vec 2.0, helping practitioners to make trade-offs when deciding on downstreak task performance.

2. We show that the most important factor for the downstream task performance is the amount of data seen during self-supervision, indicating that fixing the product of batch size and training iterations in a benchmark can provide valuable information.

3. We provide the pre-training model checkpoints with an interval of 5 k steps for further analysis.[2]

The rest of this article is structured as follows. First, we will cover related work in Section 2, including studies on batch sizes with stochastic gradient descent, (contrastive) SSL (in speech) and its scaling behaviour, and research on SSL with smaller budgets. Then, Section 3 will explain wav2vec 2.0 pre-training and fine-tuning, followed by experimental setup and results in Section 4, and we will close with a discussion and conclusion in Section 5.

---

[2]See the code repository for a link to the model checkpoints.

## 2 Related work

**Stochastic gradient descent and large batch sizes**  McCandlish et al. (2018) study the trade-off between time and computational resources when choosing a batch size. It is argued that a small batch size leads to gradients dominated by noise, which is averaged out by consecutive update steps, or more efficiently, by using data parallelism. However, when a batch size is very large, the gradient estimate contains little noise, and therefore sampling two batches and averaging their gradient will not lead to a significantly better estimate. In this case doubling the batch size does not serve a practical purpose anymore. Thus, there is a critical batch size, the exact value varying for each task and domain, after which an increase in batch size has strongly diminishing returns. Complementary, Shallue et al. (2019) conducted a study on the batch size affecting generalization performance, across multiple datasets (5 vision, 2 text), neural network families (FC, 3 CNNs, LSTM, Transformer) and optimizers (SGD, with (Nesterov) momentum). They experimentally confirm the existence of a critical batch size, and observe the magnitude of this critical batch size depends on the dataset, neural network type, and optimizer, but no clear relationship is found. With respect to the optimization trajectory, Smith et al. (2018) showed that a decaying learning rate schedule is equivalent to increasing the batch size during training, up to a batch size around 10 % of the training dataset size. There is also work on a *linear* scaling law for SGD by  Goyal et al. (2018) and *square root* scaling law for adaptive methods such as Adam by Malladi et al. (2022). These laws suggest that, for a fixed number of epochs, the same optimization trajectory can be obtained with different batch sizes by adjusting the learning rate.

**Scaling self-supervised representation learning**  Kaplan et al. (2020) analyse the scaling behavior of pre-training large language models, with respect to model size, dataset size, and the number of training steps. Their primary finding is that the test loss follows a power law in relation to all three aspects, as long as model size is increased according to the dataset size, and training length is not made a bottleneck. It is also found that very large models are more sample efficient, i.e., fewer iteration or less data is required compared to smaller models. Goyal et al. (2019) study the scaling behavior of visual representation learning, with respect to model size, dataset size, and complexity of the pretext task. They find that increasing both the dataset size, and complexity of the pretext task, is beneficial, as long as the model size is large enough. For speech SSL, the scaling behavior of the model size and the fine-tuning dataset size is studied in Pu et al. (2021). They use the reconstructive pretext task Mockingjay (Liu et al., 2020), and their results match Kaplan et al. (2020); larger model size leads to better performance, and larger models require less fine-tuning data.

**Contrastive learning and batch size**  Contrastive self-supervision benefits from large batch sizes, as ablated in SimCLR (Chen et al., 2020), and shown by, e.g., CLIP (Radford et al., 2021) and Florence (Yuan et al., 2021). A hypothesis for this observation is that distractors are often sampled within the same mini-batch, and thus more (and potentially better) distractors are available as the batch size increases. However, Mitrovic et al. (2020) show that computing the contrastive objective with fewer (e.g., only two) distractors per anchor leads to better performance, indicating that large batch sizes are the key factor of improved performance, and not the amount of available negative samples. Chen et al. (2022a) argue that small batch sizes in contrastive learning suffer from a gradient bias, which large batches sizes alleviate. Note that in wav2vec 2.0, negative samples are only taken from the same utterance. The batch size does not have any effect on the quality and quantity of negative samples, so there might be a gradient bias even with large batch sizes.

**Self-supervised learning with academic budget**  The apparent effectiveness of large batch sizes makes it difficult to do research without a large computational budget. There has been some work on trying to reduce the resources required to do pre-training. For example, Izsak et al. (2021) pre-trained a BERT model to nearly equivalent performance with only 8 GPUs (with 12 GB VRAM) in 1 day, compared to 4 days with 16 TPUs (having 32 GB RAM) in the original work (Devlin et al., 2019). This was done by reducing the maximum sequence length, focusing on large models, pre-masking data, and using specialised software packages such as DeepSpeed (Rajbhandari et al., 2020) and Apex (Micikevicius et al., 2018). Similar work has been done for the HuBERT model in Chen et al. (2023). They show that using target representations from a fine-tuned ASR model in the first iteration of HuBERT pre-training (instead of MFCCs) leads to better performance, while needing fewer GPU hours. Another line of thinking is presented by Cao & Wu (2021), where it is shown that self-supervised learning in vision can be done on small datasets, with low resolution images, and with models with relatively few parameters.

## 3 Methodology

**Wav2vec 2.0 model architecture** In this work we use the standard architectural setup for self-supervised learning with audio (Baevski et al., 2020; Hsu et al., 2021; Chen et al., 2022b). First, the raw 16 kHz sampled audio $\mathbf{X} = \{x_1, \ldots, x_r\}$ is processed with a 1-d CNN into local speech representations $\mathbf{Z} = \{\mathbf{z}_1, \ldots, \mathbf{z}_T\}$, $T = \lfloor r/320 \rfloor$. For each respective layer, the kernel sizes are [10, 3, 3, 3, 3, 2, 2] with strides [5, 2, 2, 2, 2, 2, 2], with GELU activation, 512 output channels, and GroupNorm once after the first convolution.

The CNN is followed by a gradient scaling layer with the constant set to $\frac{1}{10}$, then $\mathbf{Z}$ is projected to $\mathbf{Z}'$, with 768 dimensions, and LayerNorm is applied. Then, a relative positional embedding is added using a convolution with kernel size 128, weight normalization (Salimans & Kingma, 2016) on the kernel weights, padding of 64 on both sides, and 16 groups. A vanilla encoder-only transformer network is used to create contextualized representations $\mathbf{C} = \{\mathbf{c}_1, \ldots, \mathbf{c}_T\}$. The transformer network has 12 layers, with an hidden dimension of 768 in the self-attention module, 12 attention heads, and a scale-up to 3072 dimensions in the feed-forward network, with GELU activation.

### 3.1 Self-supervised pre-training

**Quantization** Each $\mathbf{z}_t$ in a sequence $\mathbf{Z}$ is individually classified to a quantized vector $\mathbf{q}_t$, thereby creating $\mathbf{Q} = \{\mathbf{q}_1, \ldots, \mathbf{q}_T\}$. The possible quantized vectors are learned, and represented by *codebook*s, discrete sets of real-valued vectors. A codebook $G$ is a set of $V$ entries (vectors) of a particular dimension $d_G$, representable as a matrix of size $V \times d_G$. A single linear layer with gumbel-softmax (Jang et al., 2017) activation can be used to get a probability distribution over $V$ different classes. The class with maximum probability can then be used to determine $\mathbf{q}_t$ from $\mathbf{z}_t$. In wav2vec 2.0 there are two codebooks with each $V = 320$ entries of size $d_G = 128$, resulting in $V^2 = 102\,400$ quantized vectors with dimensionality $d_q = 256$. During pre-training, a temperature $\tau = 2$ in the gumbel softmax is gradually decreased to $\tau = 0.5$ with a factor 0.999995 every iteration.

**Masking** The masking is done after the projection and normalization, but before the relative positional embedding. The mask consists of multiple regions of $L_m = 10$ consecutive latent speech vectors which are all replaced by the same learned mask vector. In total $p_m = 50\%$ of the latent vector sequence $\mathbf{Z}'$ are masked to $\hat{\mathbf{Z}}'$, with possible overlap, and excluding padding. The set of time steps where masking is applied is indicated by $\mathbf{M}$.

**Objective function** The objective function during SSL pre-training consists of a weighted sum of the main *contrastive loss* $\mathcal{L}_c$, together with an auxiliary *diversity loss* $\mathcal{L}_d$ with weighing $\lambda_d$, and an auxiliary *L2 penalty loss* $\mathcal{L}_p$ with weighting $\lambda_p$:

$$\mathcal{L}_{\text{ssl}} = \mathcal{L}_c + \lambda_d \mathcal{L}_d + \lambda_p \mathcal{L}_p \tag{1}$$

The contrastive loss $\mathcal{L}_c$ encapsulates the pretext task, where the transformer has to predict the cluster centroids $\mathbf{Q}$ of the projected and masked values $\hat{\mathbf{Z}}'$ in the output $\mathbf{C}$. The network is explicitly penalized if $\mathbf{c}_t$ is similar to any distractors sampled from $\mathbf{Q}$. Similarity is measured with the cosine similarity, written as $s(\mathbf{a}, \mathbf{b})$. The loss can then be defined as

$$\mathcal{L}_c(\mathbf{C}, \mathbf{Q}, \mathbf{M}) = \sum_{t \in \mathbf{M}} -\log \left( \frac{\exp\big(s(\mathbf{c}'_t, \mathbf{q}'_t)/\tau_c\big)}{\sum\limits_{d \in D_t \cup \{t\}} \exp\big(s(\mathbf{c}'_t, \mathbf{q}'_d)/\tau_c\big)} \right) \tag{2}$$

where $'$ implies a linear projection layer to 256 dimensions, and $D_t$ is random sample of $k = 100$ values from $\mathbf{M} \backslash \{t\}$. A temperature $\tau_c = 0.1$ leads to a hard softmax distribution. The contrastive loss can be interpreted as a standard $1 + k$ classification task with the sum-reduced cross-entropy criterion, where the target is always the class index of $\mathbf{q}_t$.

**Diversity loss** A shortcut to optimizing the contrastive loss is to map all values in $\mathbf{Z}$ to the same quantized vector. To prevent this, a diversity loss is applied, which encourages uniform predictions over the codebook

entries. A codebook $G$ with $V$ entries has classified the sequence $\mathbf{Z}$ to $\mathbf{Q}$, using logits from a softmax activations of a linear layer $\mathbf{P} = \{\mathbf{p}_1, \ldots, \mathbf{p}_T\}$. For uniform predictions the average probability distribution $\bar{\mathbf{p}} = T^{-1} \sum_{t=1}^{T} \mathbf{p}_t$ should be flat. In this *best* case, the entropy $H(\bar{\mathbf{p}}) = \log V$, and the perplexity $e^{H(\bar{\mathbf{p}})} = V$. In the case of *shortcut*, a single class has probability 1, which means the entropy $H(\bar{\mathbf{p}}) = 0$ and the perplexity 1. Therefore, the diversity loss minimizes the number of the entries in a codebook subtracted by the perplexity of the predictions:

$$\mathcal{L}_d(\mathbf{P}) = V - \exp\big(-\sum_{j=1}^{S} \bar{p}^{(j)} \log \bar{p}^{(j)}\big), \tag{3}$$

where $\bar{p}^{(j)}$ is the $j$th component of $\bar{\mathbf{p}}$. The weighting $\lambda_d = \frac{1}{10}$ throughout this work unless specified otherwise.

**L2 penalty loss** The third loss is a regularization term, which keeps the values of $\mathbf{Z}$ as small as possible. This loss is defined as

$$\mathcal{L}_p(\mathbf{Z}) = \frac{1}{Td_z} \sum_{t=1}^{T} \sum_{j=1}^{d_z} \big(z_t^{(j)}\big)^2, \tag{4}$$

where $z_t^{(j)}$ is the $j$th component of $\mathbf{z}_t$, and $d_z = 512$. The weighting $\lambda_p = 10$ throughout this work.

## 3.2 Batch creation

The methodology descriptions have assumed a single utterance $\mathbf{X}$, while training is done with a batch of the dataset, split into multiple gpu-batches for distributed data-parallel training. The LibriSpeech dataset is used, implying utterances have variable lengths, at minimum 0.83 seconds, and at most 30 seconds. As each utterance in a gpu-batch needs to have the same length, all but the longest raw waveform in a batch are padded with zeros. To minimize the amount of padding, the utterances are sorted by length, and put into bins of 5000 utterances. Each gpu-batch is sampled from only a single bin. Random samples from the bin feed a priority queue of length 50, from which gpu-batches are formed by taking samples prioritized by shortest duration, until the total speech duration in the gpu-batch exceeds a threshold, in our case 2.4 M samples. Because of limitation in GPU memory caching, a gpu-batch is discarded if the difference between the shortest and longest utterance is more than 10 seconds. This helped alleviate spontaneous CUDA out-of-memory errors.

The CNN also processes the padded part of utterances. However, every vector $\mathbf{z}_t$ which results purely from padding in the raw waveform are ignored in the self-attention of the transformer by setting their attention score to $-\infty$. When creating the mask $\mathbf{M}$, the padded vectors $\mathbf{z}_t$ are also not considered part of the utterance. The contrastive loss is computed independently for each masked token in each utterance of the gpu-batch, and summed afterwards. For the diversity loss, the probability distribution $\bar{\mathbf{p}}$ is computed by averaging over the predictions of all tokens of all utterances in the gpu-batch, before computing the perplexity. The L2-penalty loss is simply the mean of each representation value in the gpu-batch. The gradient resulting from each gpu-batch are averaged before the weights of the networks are updated.

## 3.3 Full fine-tuning for speech recognition with subsets of LibriSpeech

To fine-tune a pre-trained model for speech recognition, $\mathbf{Z}$ and $\mathbf{C}$ can be computed from $\mathbf{X}$, disregarding the quantization. The network still applies a mask, but only $p_m = 5\%$ of the utterance is replaced with the learned masking vector. This acts as a regularization method, similar to SpecAugment (Park et al., 2019). Each vector in $\mathbf{C}$ can be separately classified to a character (or blank) with a softmax-activated linear layer, and optimized with CTC (Graves et al., 2006) loss. The CNN is not updated during fine-tuning, and the transformer network is only updated after the first 5 k iterations. We fine-tune on 10 min, 1 hour, 10 hours, 100 hours and 960 hours of LibriSpeech following Baevski et al. (2020).

## 3.4 Frozen fine-tuning for various speech technology tasks using the SUPERB benchmark

Another fine-tuning strategy is used in the SUPERB benchmark, where the (upstream) CNN and transformer layers will be frozen and used only to generate input features for a task-dependent, small downstream model,

Table 1: All batch sizes used for SSL pre-training, together with the number of GPUs, the number of gradient accumulation steps (acc), the runtime, in days and hours, of a single run, and three possible learning rate heuristics. The bold learning rates resulted in the lowest validation loss. For the 80 minute batch size setting we only tried one learning rate.

| batch size | | GPUs | acc. | runtime | used learning rates | | |
|---|---|---|---|---|---|---|---|
| sec | min | | | | $h_{\text{const}}$ | $h_{\text{sqrt}}$ | $h_{\text{lin}}$ |
| 87.5 | 1.5 | 1 | 1 | 1d 13h | $5.00 \times 10^{-4}$ | $\mathbf{6.04 \times 10^{-5}}$ | $7.29 \times 10^{-6}$ |
| 150 | 2.5 | 1 | 1 | 2d 4h | $5.00 \times 10^{-4}$ | $\mathbf{7.91 \times 10^{-5}}$ | $1.25 \times 10^{-5}$ |
| 300 | 5 | 1 | 2 | 3d 22h | $5.00 \times 10^{-4}$ | $\mathbf{1.12 \times 10^{-4}}$ | $2.50 \times 10^{-5}$ |
| 600 | 10 | 4 | 1 | 2d 14h | $5.00 \times 10^{-4}$ | $\mathbf{1.58 \times 10^{-4}}$ | $5.00 \times 10^{-5}$ |
| 1200 | 20 | 2 | 4 | 7d 20h | $5.00 \times 10^{-4}$ | $\mathbf{2.24 \times 10^{-4}}$ | $1.00 \times 10^{-4}$ |
| 2400 | 40 | 4 | 4 | 7d 18h | $\mathbf{5.00 \times 10^{-4}}$ | $3.16 \times 10^{-4}$ | $2.00 \times 10^{-4}$ |
| 4800 | 80 | 8 | 4 | 7d 19h | $\mathbf{5.00 \times 10^{-4}}$ | - | - |

e.g., a 2-layer biLSTM for speech recognition, or a single dense layer followed by mean pooling for speaker identification. The input features $\mathbf{F} = \{f_1, ..., f_T\}$ are a weighted-sum of $\mathbf{Z}'$ and $\mathbf{C}$, i.e., $f_t = w_0 z_t' + \sum_{i=1}^{12} w_i c_t^{(i)}$, where $^{(i)}$ indicates the $i$th transformer layer output sequence, and $w_i$ is learned during fine-tuning. We use a subset of tasks to limit computational resources, while including at least one task from 4 out of the 5 categories: phoneme recognition (content), ASR in English (content), out-of-distribution ASR in Mandarin (content), speaker verification (speaker), emotion recognition (prosody), and intent classification (semantics). We use the default downstream model for each task, and keep most settings to the default value. We make the following modifications to reduce the run-time: half- instead of single-precision floats, 200k instead of 500k train steps for mandarin ASR, and gradient accumulation of 1 for all tasks, while increasing the batch size such that the effective batch size is equal to the default settings. A learning rate of $10^{-4}$ is used for all tasks.

## 4 Experiments

### 4.1 Pre-training with different batch sizes

The first experiment aims to directly answer RQ 1, and is a prerequisite for answering all others.

**Setup** We pre-train the BASE wav2vec 2.0 network with batch sizes ranging from 87.5 seconds to 80 minutes of audio, as seen in Table 1. Each pre-training starts with the same initial weights, and we use all 960 hours of training data in LibriSpeech (Panayotov et al., 2015) (CC-BY 4.0 licence), with 5% held-out randomly as a validation set. We validate and store a checkpoint every 5 k steps. We adhere to the hyperparameters as published in the seminal paper (Baevski et al., 2020) as much as possible. We use 400 k training iterations with AdamW and a weight decay of $10^{-2}$, and scan over 3 learning rates (LRs), based on choices explained below. We change to a 8-cycle triangular learning rate schedule, where one cycle has 25 k linear steps up and 25 k linear steps down. This allows us to fairly compare fine-tuning results of checkpoints at multiples of 50 k iterations. The minimum LR of the cycle is 100 times smaller than the maximum LR. We show the maximum LRs in Table 1. We also use GPUs (A5000) with at least 24 GB of VRAM , and therefore fill each GPU with a maximum of 2.4 M audio samples (150 seconds) for full utilization of the device, which compares to 1.4 M samples (87.5 seconds, 90 minutes batch size with 64 GPUs) in Baevski et al. (2020). The experiments, including initial development runs, took 246 days of GPU time.

For each batch size of duration $s$, we need to find a well-performing maximum learning rate (LR) for the cyclic schedule. As a full hyperparameter search would exceed our computational budget, we use heuristics to choose three different learning rates, and settle on a run with the lowest overall validation loss. The first heuristic is the linear scaling law from Goyal et al. (2018). As a reference, a maximum learning rate of $m_{\text{lr}} = 5 \times 10^{-4}$ was used in Baevski et al. (2020) together with a batch size of circa 1.6 hours. Therefore we use $h_{\text{lin}}(s) = m_{\text{lr}} s / s_{\text{orig}}$ as the first heuristic for the learning rate, with $s_{\text{orig}} = 6000$ seconds. In Baevski et al. (2020) a batch size of 5600 seconds is used, but we use 6000 seconds for this heuristic calculation so that

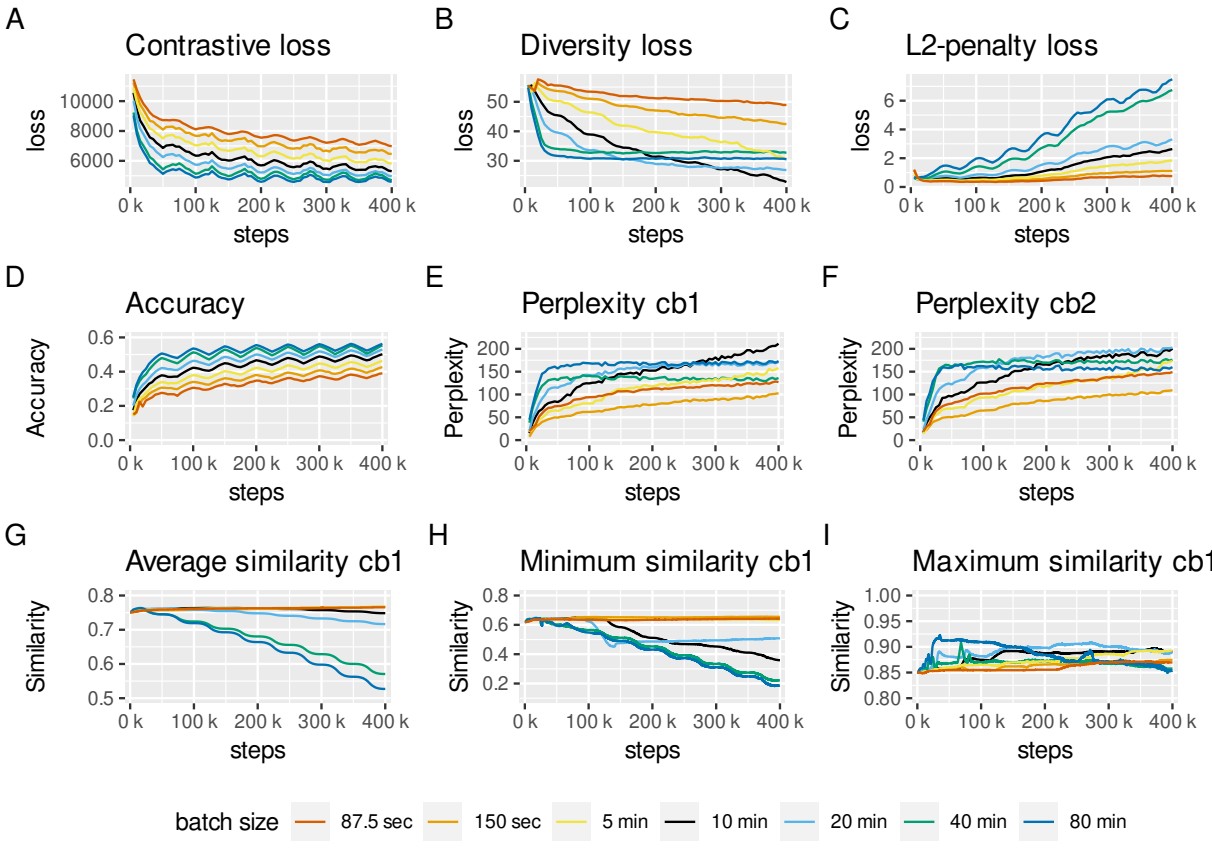

Figure 1: Various metrics on validation data (interval of 5 k training steps) during self-supervised pre-training with different batch sizes, namely all three losses (A, B, C), the accuracy of predicting the correct masked quantized vector (D), and the perplexity of codebook 1 (E) and codebook 2 (F). We also show the average, minimum, and maximum value of the pair-wise cosine similarity of all codewords in codebook 1 (G, H, I) with an interval of 100 training steps.

$h_{\text{lin}}$ rounds nicely. We still find well-performing LRs. The second heuristic $h_{\text{sqrt}}(s) = m_{\text{lr}} \sqrt{s/s_{\text{orig}}}$ applies the square root scaling law according to Malladi et al. (2022). For each batch size we also try the constant $h_{\text{const}}(s) = m_{\text{lr}}$, although this led to divergence for $s \leq 600$ seconds. Moreover, we initially used a diversity loss weight $\lambda_d = 0.1$ as in Baevski et al. (2020), but found this led to divergence of the diversity loss for the batch size of 87.5 seconds. By decreasing $\lambda_d$ from 0.1 to 0.05, with the reasoning that the contrastive loss is almost twice as small (only 1.4 M samples instead of 2.4 M), and the ratio between the contrastive loss and the diversity loss should stay the same, we managed to get converging results for the batch size of 87.5 seconds. This does seem counter-intuitive as the original work uses 1.4 M samples on each of the 64 GPUs together with $\lambda_d = 0.1$.

**Results** We show various metrics during the training procedure in Figure 1. For each batch size, the metrics of the run with the lowest validation loss are displayed. For the contrastive loss (A), we see that overall a *larger batch size leads to a lower contrastive loss*. Note that the smallest batch size, 87.5 seconds, has a different range because the loss is sum-reduced over 1.4 M sampled instead of 2.4 M samples, we corrected for this by multiplying the values by 1.71. For the diversity loss (B), we see that a large batch size (40, 80 min) causes the loss to drop quickly, but then plateau. The other batch sizes steadily decrease. Notably, for batch sizes of 10 and 20 minutes the diversity loss surpassed the values of batch sizes 40 and 80 minutes after 150 k to 200 k steps. The scale of the lowest batch size, 87.5 seconds, is twice as small due to the $\lambda_d = 0.05$, we corrected for this by multiplying the values by 2. The same patterns visible in the diversity loss are also

seen in the perplexity of the codebooks (H and I, with the vertical scale reversed). For accuracy (D), a larger batch size leads to higher accuracies. For the similarity of codewords within the codebooks, we see that the average (G) and minimum (I) *cosine similarity between codebooks only go down steeply with large batch sizes* (40 and 80 min). For the maximum similarity values, we observe that they stay relatively stable, although for the larger batch sizes they increase slightly at the start of training, but decrease again during the training procedure, which can be related to the decay strategy of $\tau$ used in the gumbel-softmax.

## 4.2 ASR fine-tuning with varying amounts of labels

The second experiment focuses on RQ 2. How is downstream fine-tuning affected by the batch size during pre-training?

**Setup**  For each batch size in Table 1 we have self-supervised training runs of 400 k steps, with checkpoints saved every 5 k steps. For each setting we select the run (and step) with the lowest overall validation loss, resulting in a single checkpoint which is used as initialization for training a speech recognition system. This is the checkpoint at step 400 k for all runs, expect for batch size of 32 GPUs (80 minutes), which had the lowest validation loss at step 305 k. For each of these selected checkpoints we perform a fine-tuning on 10 minutes, 1 hour, 10 hours, 100 hours, and 960h hours of labeled LibriSpeech data, with hyperparameters detailed in Appendix A.3. We use the same number of steps as in Baevski et al. (2020). We show results with greedy letter decoding and word decoding using a 4-gram LibriSpeech language model. For word decoding we use use a beam size and threshold of 50, a language model weight of 2, and a word insertion score of 0 for all settings. These experiments were done using one A5000 GPU, with a maximum run-time of 2 days when fine-tuning with 960h of labels (320k steps). In total 205 days of GPU time was used, including experiments in section 4.3.2.

**Results**  We show the word-error-rate (WER), evaluated on LibriSpeech test-clean and test-other, for each fine-tuning condition in Figure 2 and in tabular format in Appendix A.4. Two clear patterns are visible. First, independent of the amount of labels available, we observe that *fine-tuning a random initialization leads to the highest WER*. Then, *each consecutive increase in the batch size during self-supervised learning leads to lower WERs after fine-tuning.* There is one exception: on test-other, the 40 min batch size initialization performs better than the 80 min batch size initialization, but only when fine-tuning with 10 or more hours of labeled data. We observed similar degraded performance after fine-tuning the 400 k checkpoint with batch size of 80 minutes (not shown in Figure 2). Secondly, having more labeled data for fine-tuning leads to a lower WER for each self-supervised batch size. However, the larger the batch size, the smaller the difference in WER between the amount of labels available during fine-tuning. Notably, Baevski et al. (2020) reports 9%/47% WER on test-clean with/without a language model when fine-tuning with 10 minutes of labeled audio. In this experiment we observe a WER of 24%/41% instead, the large difference in LM performance we attribute to our much smaller beam size in decoding. Finally, we see diminishing returns at a batch size of 80 min.

## 4.3 Analysis on effectiveness of large batch sizes

So far, we have observed that larger batch sizes lead to a lower contrastive validation loss, and less similarity between codewords. We have also seen that larger batch result in better fine-tuning performance for speech recognition, irrespective of the amount of labels available. Why are large batch sizes more effective? Are better gradients approximations beneficial, or is the amount of observed data an explaining factor?

### 4.3.1 Variance of gradients

First, we compare the gradient between different batch sizes. If the gradients are more precise, and less noisy, with increased batch sizes, we expect the variance between gradients to decrease. To verify this we use the saved checkpoints (every 5 k steps) during pre-training. For each checkpoint, we compute 10 gradient vectors using new, independently sampled batches from the training set. These 10 batches are kept constant over all checkpoints of the same batch size. We do not update the weights during this process, and we do not use the AdamW optimizer state nor the learning rate to scale the gradients. For each parameter, we separately compute the standard deviation between gradient vectors, after which we average over the parameters to a get a standard deviation of the whole gradient, shown in Figure 3. *We observe that the*

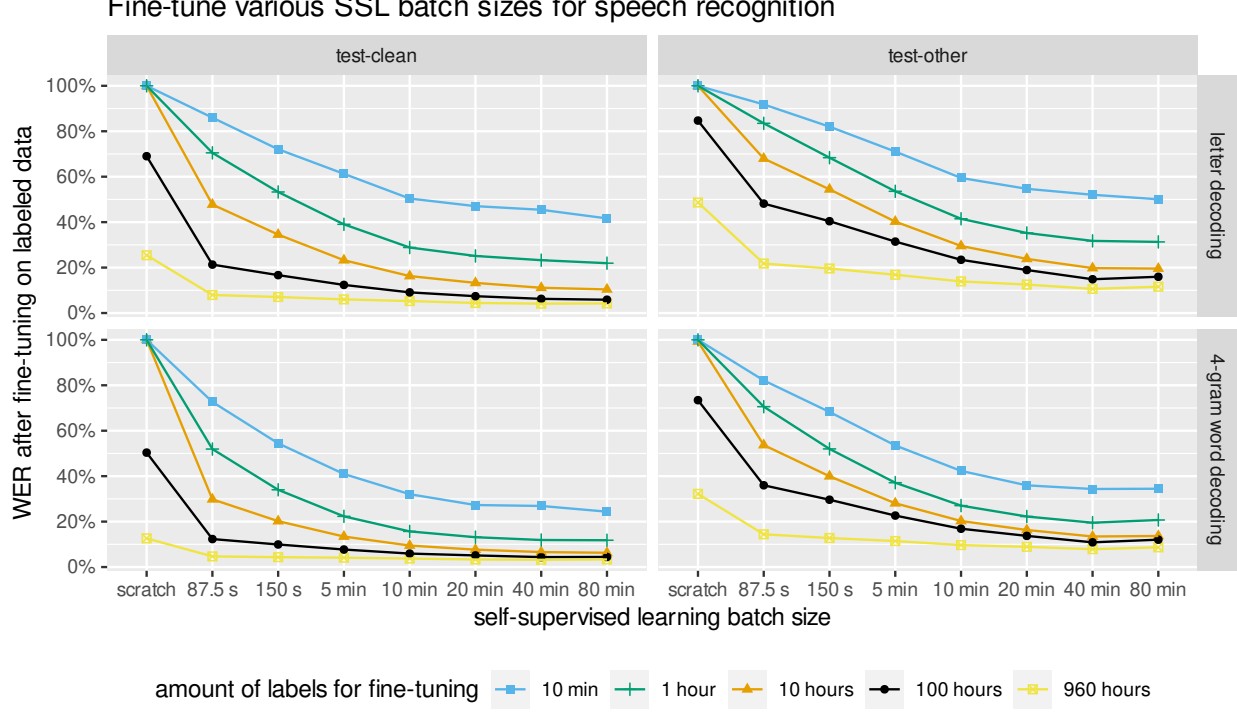

Figure 2: The WER (left column: LibriSpeech test-clean, right column: LibriSpeech test-other) against the batch size during pre-training of a self-supervised initialization. The self-supervised models are fine-tuned for speech recognition using 5 different magnitudes of labeled data. Scratch indicates fine-tuning a random initialization instead of a self-supervised initialization. The upper row shows the WER with letter decoding, while the bottom row shows the WER with word decoding using a 4-gram language model.

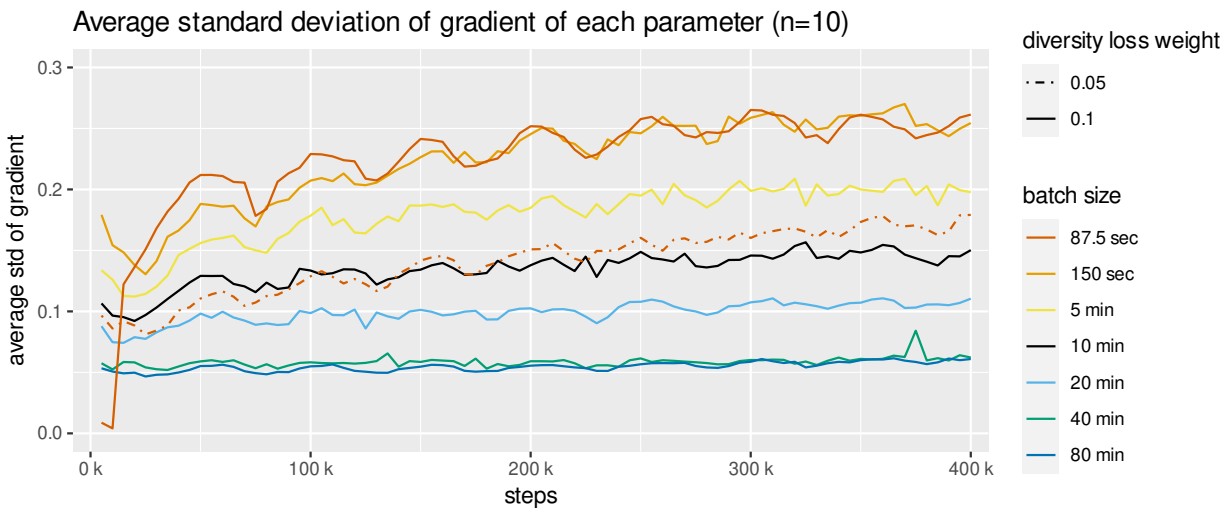

Figure 3: The standard deviation of the gradient of $\mathcal{L}_{ssl}$ for each batch size, computed over 10 random batches, and averaged over all parameters, against consecutive checkpoints during pre-training.

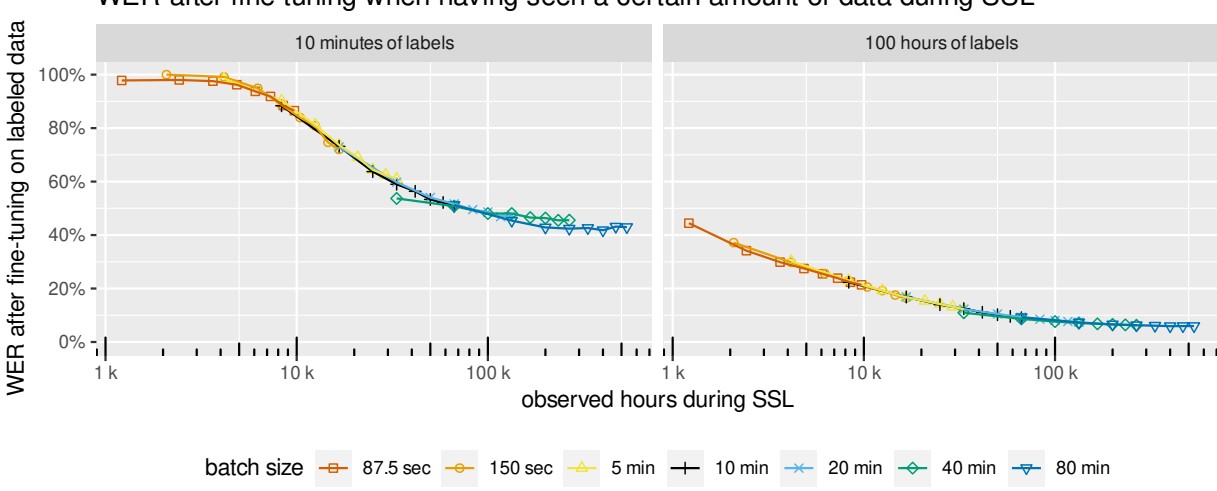

Figure 4: The WER, after fine-tuning, against the hours of data processed during self-supervision.

*standard deviation decreases as the batch size increases.* For the smallest batch size, the training run with $\lambda_d = \frac{1}{20}$ has a significantly lower gradient variance compared to $\lambda_d = \frac{1}{10}$. Further, at a critical batch size of 40 minutes the standard deviation barely decreases when the batch size is doubled to 80 minutes. Furthermore, for small batch sizes the standard deviation increases over the training procedure, while it stays constant for large batch sizes. Moreover, we observe that the batch sizes of 87.5 (with $\lambda_d = \frac{1}{10}$) and 150 seconds converge at the end of training. Finally, the cyclic learning rate schedule seems to affect the gradient variance, as the standard deviation peeks for all batch sizes when the cycle is at the minimum learning rate (every interval of 50k steps).

### 4.3.2   Fine-tuning after observing specific amounts of data during pre-training

We will now focus on RQ 3. We compare the performance of batch sizes at different stages during pre-training. Because we use a cyclic learning rate, there are equivalences at each end of a cycle, namely at multiples of 50 k steps. Different batch sizes overlap on the amount data seen at particular checkpoints. For example, 16.7 k hours of data was observed with batch sizes of 150 sec, 5 min, 10 min, and 20 min respectively at 400 k, 200 k, 100 k, and 50 k steps. If less noisy gradient approximations are beneficial to learning, we expect a performance difference when we compare the fine-tuning performance of these checkpoints, in favor of larger batch sizes. However, if all that matters is observing more data, we should see no difference in performance.

**Setup and results**   For each batch size, we fine-tune the checkpoints with an interval of 50 k steps, resulting in 8 checkpoints per batch size. We use the training methodology as described in Section 4.2. For this experiment we focus on fine-tuning on 10 minutes and 100 hours of labeled data, with letter decoding, evaluating on the test-clean set. The results are shown in Figure 4 and in tabular form in Appendix A.5. *We observe a direct relationship between the amount data observed during pre-traing and the WER after fine-tuning.* There are only minor differences between the WER of different checkpoints with the same amount of data, which we attribute to noise. The curves for each batch size blend into each other, especially for the case of fine-tuning with 100 hours of data. With 10 minutes of data we observe that a batch size of 40 minutes has slightly better performance at the start of training, and worse performance at the end of training, compared to batches of 20 minutes and 80 minutes. Also, we see that a batch size of 87.5 seconds performs slightly better than the batch size of 150 seconds. We hypothesise that the diversity weight $\lambda_d = \frac{1}{20}$ generalized slightly better than $\lambda_d = \frac{1}{10}$. Note that we used the naive, upper bound of the amounts of data hours observed. The measured amount of hours are shown in Appendix A.1.

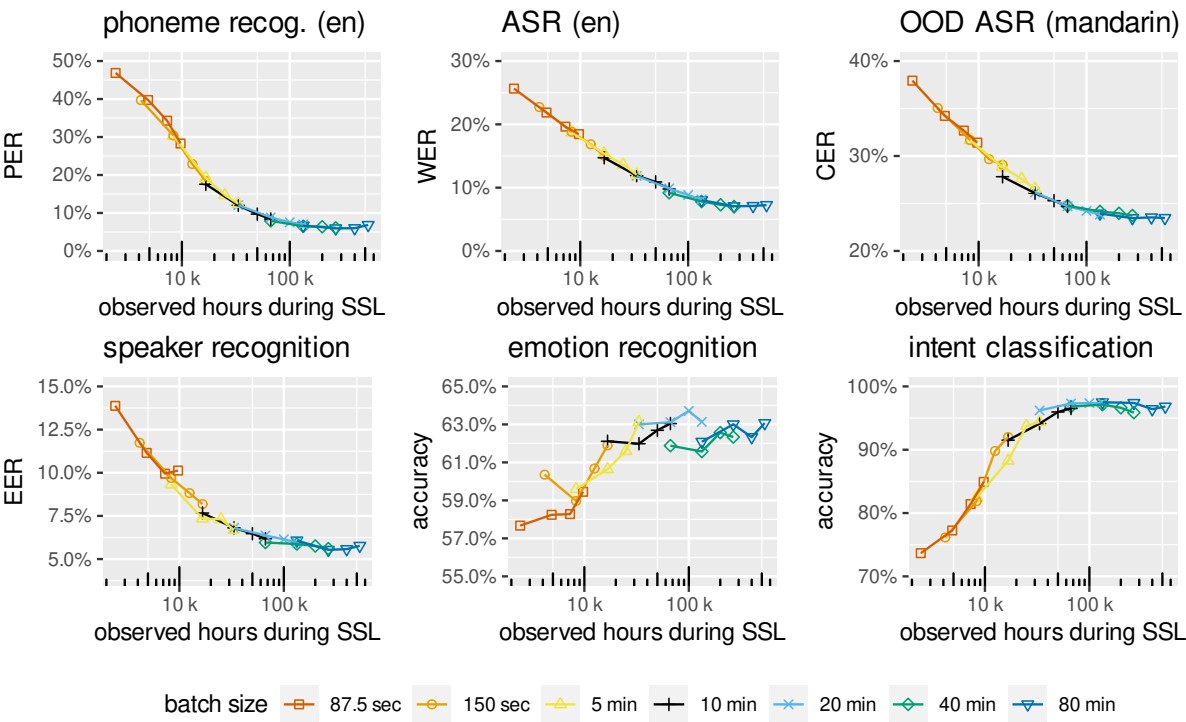

Figure 5: The performance of 6 SUPERB tasks against the hours of data processed during self-supervision. For each SSL batch size we fine-tune the checkpoints at step 100k, 200k, 300k and 400k with learning rate $10^{-4}$.

## 4.4 SUPERB fine-tuning

To further strengthen the observation that the amount of hours seen during pre-training is the main indicator of downstream task performance, we repeat the experiment in Section 4.3.2 on 6 tasks in the SUPERB benchmark (Yang et al., 2021), namely phoneme recognition, speech recognition in English and Mandarin, speaker verification, emotion recognition, and intent classification. There are 3 important differences. First, we are interested to see whether the pattern holds for tasks other than (English) speech recognition. Secondly, the speech representation features are frozen during fine-tuning, so the quality of the representations are more fairly judged. Thirdly, other than phoneme recognition and English ASR, these tasks fine-tune on out-of-domain data with respect to the pre-training, so that we can see whether the observation holds in cross-domain adaptation. Note that following datasets are used: phoneme recognition and English ASR are fine-tuned on the train-clean-100h set of LibriSpeech (Panayotov et al., 2015), mandarin ASR is fine-tuned on Mozilla Common Voice (Ardila et al., 2020), speaker verification is fine-tuned on VoxCeleb1 (Nagrani et al., 2017), intent classification is fine-tuned on Fluent speech commands (Lugosch et al., 2019), and emotion recognition is fine-tuned on IEMOCAP (Busso et al., 2008).

**Setup and results** For each of the 6 tasks, we fine-tune with 4 checkpoints of each SSL batch size, namely the checkpoint at 100k, 200k, 300k and 400k steps. We use a learning rate of $10^{-4}$ for each fine-tuning, and keep all other hyperparameters equal to the default value for the task, expect that mandarin ASR is fine-tuned for 200k steps instead of 500k steps. We show the results in Figure 5, and in tabular format in Appendix A.6. In general, *we observe similar patterns compared to Figure 4*, meaning curves blending into each other, and following the pattern of better performance after seeing more data. We also see signs of overfitting with the batch size of 80 minutes at 400k steps. We observe that the in-domain tasks (English ASR, and phoneme recognition) have the smoothest curve, followed by mandarin ASR. The other 3 tasks seem more noisy, where sometimes there is no improvement following a consecutive checkpoint of the same batch size, e.g., this is visible for speaker and emotion recognition at 10 k hours, and intent classification

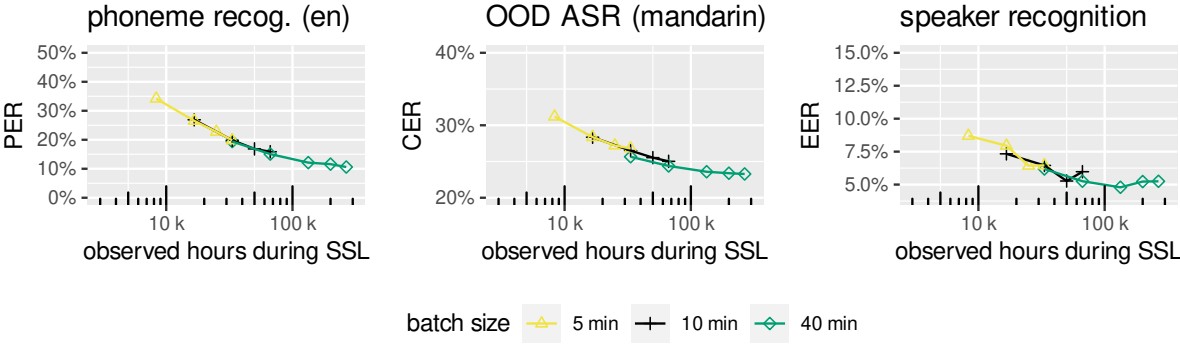

Figure 6: The performance of three SUPERB tasks against the hours of data processed during self-supervision while pre-training on the VoxCeleb2 dataset. Each checkpoint is fine-tuned with LR $10^{-4}$.

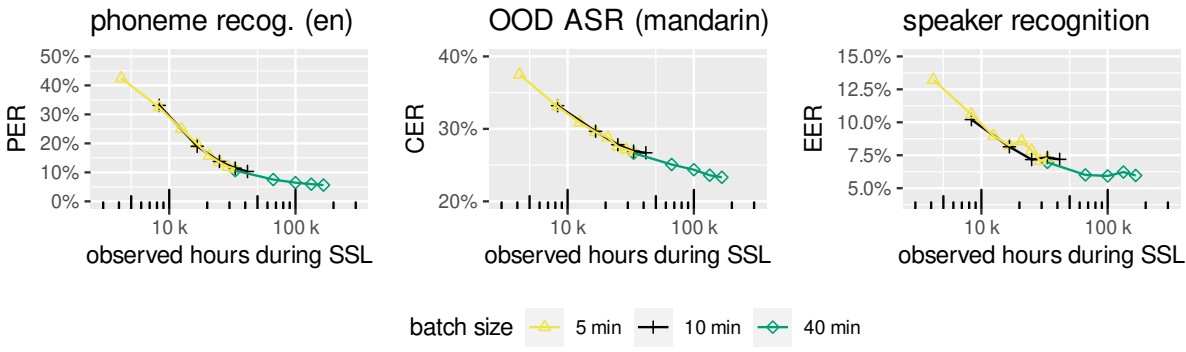

Figure 7: The performance of three SUPERB tasks against the hours of data processed during self-supervision with the LARGE model. Each checkpoint is fine-tuned with LR $10^{-4}$.

at 30 k hours. Also, emotion recognition is the only task where there is no clear upward trend with more observed data. For emotion recognition, the best performance was obtained with the checkpoint at 300 k steps with the batch size of 20 minutes, and the batch sizes of 40 and 80 minutes perform noticeably worse.

### 4.5   Increasing model capacity or changing pre-training dataset

All experiments so far have used the BASE wav2vec 2.0 network alongside pre-training on Librispeech 960h. In this section we will verify whether our observations generalize to a larger model capacity or a different pre-training dataset.

**Setup**   We pre-train with batch sizes of 5 minutes, 10 minutes, and 40 minutes of audio. For BASE, we use VoxCeleb2 (Chung et al., 2018), which differs from Librispeech in that it is primarily designed for speaker recognition and created by automatically extracting utterances from YouTube videos. Compared to Librispeech it has more speakers (2338 versus 5994) and high intra-speaker variability. We split the VoxCeleb2 development set into a train and validation partition. The train partition has approximately 1 M utterances, with a duration ranging from 4 to 30 seconds averaging at 7.8 seconds, and a total of 2300 hours of audio. The validation partition is created by randomly selecting 2 utterances from each speaker session, and has a total of 23 hours of data. For pre-training we do not make any changes to the network or optimization procedure detailed in Subsection 4.1, other than performing a learning rate scan. For each batch size we perform a grid search in the range $10^{-4}$ to $7 \times 10^{-4}$ with increments of $5 \times 10^{-5}$.

For LARGE with Librispeech, we change the network to use 24 transformer layers, with 16 heads, and a hidden dimensionality of 1024 which is scaled up to 4096 in the feed-forward network. Furthermore, the

codeword dimensionality is increased from 128 to 384, hence the dimensionality of the projection $s(\mathbf{c}'_t, \mathbf{q}'_d)$ in the contrastive loss is increased from 256 to 768. These modifications increase the amount of parameters from 95 M to 316 M. Finally, the floor of the gumbel softmax temperature of the diversity loss is decreased from 0.5 to 0.1. We also changed the diversity loss weighting from $\lambda_d = 0.1$ to $\lambda_d = 0.01$, because we observed that for all learning rates $\lambda_d = 0.1$ resulted in the diversity loss immediately converging and the contrastive loss never improving. We trained for 400 k steps with a batch size of 5 minutes and 250 k steps for batch sizes of 10 and 40 minutes. As in Subsection 4.1, we performed a learning rate scan for each batch size using the 3 heuristics $h_{\text{const}}$, $h_{\text{sqrt}}$, and $h_{\text{lin}}$. We used $m_{\text{lr}} = 3 \times 10^{-4}$ and $s_{\text{orig}} = 9600$ seconds taken from Baevski et al. (2020).

**Results**  For BASE with VoxCeleb2, we found the best performing LR for batch size 5, 10 and 40 minutes to be respectively $2 \times 10^{-4}$, $3 \times 10^{-4}$ and $5 \times 10^{-4}$, and for LARGE with Librispeech respectively $5.3 \times 10^{-5}$, $7.5 \times 10^{-5}$ and $1.5 \times 10^{-4}$. We fine-tune consecutive checkpoints (50 k steps for LARGE, and 100 k steps for VoxCeleb2) on 3 SUPERB tasks, namely English phoneme recognition on Librispeech, Mandarin speech recognition on CommonVoice, and speaker recognition on VoxCeleb1. Each checkpoint is fine-tuned with learning rate $10^{-4}$. We show the results with VoxCeleb2 as pre-training data in Figure 6 and for LARGE in Figure 7. *For both settings we confirm the general trend of better performance as the visible training hours during pre-training increases.* We observe for Mandarin speech recognition, and pre-training with VoxCeleb2, that the batch size of 40 minutes has slightly better performance across the amount of data observed, compared to the batch sizes of 5 and 10 minutes. For the speaker recognition task, we see for both VoxCeleb2 and LARGE that the performance does not monotonically decrease with the amount of hours of observed data within the optimization trajectory of a single batch size. This is consistent with observations in Figure 5. We see better EERs for speaker recognition with the BASE model pre-trained on in-domain data (VoxCeleb2) compared to the BASE and LARGE model pre-trained on out-of-domain data (Librispeech). Finally, for the phoneme recognition task, we see, for both conditions, the least amount of noise in the trend of performance versus amount of data seen.

## 5  Discussion and conclusions

**Research questions**  From the extensive search of batch sizes reported in Figure 1, we see that larger batch sizes result in better pre-training convergence, if given the same amount of iteration. This is consistent with the hypothesis of RQ 1 and 2. We were surprised to observe convergence with all batch size, with the caveat that we had to change the diversity loss weighting for the smallest batch size of 87.5 seconds. It seems this problem with the loss weighting parameter choice could have been prevented if the contrastive loss would be mean-reduced instead of sum-reduced, but in this work we followed the implementation of Baevski et al. (2020) as closely as possible. A good indicator for well chosen hyperparameters is a continuous increase of the perplexity of the codebook logits (Figure 1E–F). With larger batch sizes, the similarity of codebook vectors decreases, which is an indication of the diversity of the learnt representations.

Regarding RQ 2 and Figure 2, we show, for the first time, how pre-training batch size affects the downstream ASR performance: with a fixed number of iterations, the performance increases with larger batch size. All results with the 80 min batch size are in accordance with the original paper Baevski et al. (2020), except for the 10 minute fine-tuning results, where they decoded using the (impractical) beam size of 500. Our cyclic LR schedule does not perform worse than Baevski et al. (2020), while allowing for a fair comparison when conducting fine-tuning experiments at regular intervals. The largest batch size we investigated showed a little worse performance. We might have reached a limit of the generalization ability of the pretext task, as corroborated by the minimum validation loss at 305 k steps. We expect this limit can be increased by more regularization, e.g., using a larger data set (Conneau et al., 2021; Radford et al., 2023), or, to a lesser extent, higher dropout rates or higher value of $\lambda_p$ and $p_m$.

In looking for an answer to why larger batch sizes are more effective, we saw in Figure 3 that the standard deviation of the gradients reduces almost consistently with larger batch size, up to a value of 40 min. However, we do not see an indication that this has an effect on downstream task performance, cf. Figure 4 and 5. The reduced variance in the gradient as a result of a larger batch size does not appear to be the reason for better performance. Rather, we found that the most important factor for downstream task performance is the total amount of data seen during pre-training, i.e., the product of batch size and number of iterations (RQ 3), as

shown convincingly in Figure 4 and Figure 5 as well as Figure 6 and Figure 7. This means that it still is possible to carry out pre-training with limited amount of GPUs and/or memory, but one needs to be more patient or accept a penalty in performance, where Figure 2 can help in decision making.

For the non-content tasks in Figure 5, namely speaker, emotion and intent recognition, there appears to be more noise in relation between amount of data seen and performance. This noise is strongest for emotion recognition, however, the results lie in a 10 % accuracy bandwidth, which is similar to the range of worst to best systems in the SUPERB leaderboard (Yang et al., 2021). The small size of the IEMOCAP dataset (10 speakers with 5-fold cross validation) probably adds to the noisy behaviour. We believe this task may not be the best to indicate the quality of the learned speech representations. When the pre-training dataset is out-of-domain with respect to the fine-tuning dataset, we also see more noisy behaviour, cf. Figure 6, however, the general trend still is visible. Finally, we see that the trend also holds when we increase the model capacity, cf. Figure 7.

**Broader impact**   Based on these results, we believe that it could benefit the community to benchmark SSL algorithms (in speech) by constraining the amount of data seen in training, e.g., to 100 k hours. In experiments with different algorithms, one might use 10 k hours of seen data to reduce the computational burden, and verify conclusions at the 100 k hours pre-training condition.

**Conclusion and limitations**   We conclude that the batch size during contrastive pre-training can be varied over a large range of values without a performance penalty, but only if hyperparameters like the learning rate are adapted accordingly. As a caveat, our results have only looked at contrastive algorithms where distractors are *not* taken from other samples in the batch. Future work could look at algorithms where this *is* the case, such as SimCLR, or a modified wav2ec 2.0, paying special attention to the square root scaling law of the learning rate (Malladi et al., 2022) such that all batch sizes have similar well-performing optimization trajectories. Moreover, other speech representation learning algorithms could be considered, such as predictive methods like HuBERT (Hsu et al., 2021), WavLM (Chen et al., 2022b) and DinoSR (Liu et al., 2023), and reconstructive methods like VQ-VAE (Oord et al., 2017) and DeCoAR (Ling et al., 2020). The range of the batch sizes we found effective for wav2vec 2.0 may be specific to the architecture (wav2vec 2.0 base, with approximately 95 M parameters) and our pre-training datasets, but we have shown that also a larger model, and a different pre-training dataset, shows a dependence on the amount of data seen, as shown in Figure 6 and Figure 7, in terms of the fine-tuning performance.

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

# A  Appendix

## A.1  Measured hours of observed data during pre-training

Table 2: The number of epochs and total amount of data observed throughout pre-training for each batch size. The training dataset contains 912 hours of data and we train for 400 k iterations. Note that the batch size is an upper bound as they are constructed with variable length samples.

| batch size | | upper bound | | measured | |
|---|---|---|---|---|---|
| sec | min | epochs | observed data (h) | max. repeats of sample | observed data (h) |
| 87.5 | 1.5 | 11 | 10 k | 11 | 9 k |
| 150 | 2.5 | 18 | 17 k | 18 | 16 k |
| 300 | 5 | 37 | 33 k | 36 | 31 k |
| 600 | 10 | 73 | 67 k | 71 | 62 k |
| 1200 | 20 | 146 | 133 k | 140 | 124 k |
| 2400 | 40 | 292 | 267 k | 277 | 248 k |
| 4800 | 80 | 585 | 533 k | 554 | 497 k |

Due to fact that audio samples in LibriSpeech are varied, batch sizes are filled up to a specific threshold, as explained in Section 3.2. The product of the batch size and number of iterations is therefore an upper bound on the amount and duration of samples observed. Moreover, we discard some batches when the difference between the minimum and maximum file is larger than 10 seconds, as this prevented GPU out-of-memory errors. During pre-training, we stored the identifiers of every utterance in all 400 k batches. Afterwards, we could compute the actual amount of data observed, looking up the length of each utterance without padding. The results are shown in Table 2. We can see that the largest batch size has actually seen only 497k hours of data, instead of the theoretical 533 k hours. This shows the importance of creating batches with as little length variability as possible, because our results have shown that seeing more hours of data in the same amount of iterations matters for downstream performance.

## A.2  Pre-training plots with visible hours as x-axis

Figure 8, Figure 9, Figure 10, Figure 11, Figure 12 and Figure 13 show the plots of Figure 1 with the amount of data seen as x-axis instead of the amount of iterations. To correct for the different gpu-batch sizes (1.4 M samples instead of 2.4 M samples) and diversity loss weighting (0.05 instead of 0.10), we multiple the values of batch size 87.5 seconds with a factor 1.71 for the constrastive loss and 2 for the diversity loss.

## A.3  Hyperparameter details for full fine-tuning

We use similar fine-tuning parameters for each labeled data condition as stated in Baevski et al. (2020), but make some changes such that the only variation is in the number of iterations. We use 12 k, 13 k, 20 k, 80 k and 320 k iterations, respectively for 10 minutes, 1 hour, 10 hours, 100 hours, and 960h hours of labeled fine-tuning data. We use a learning rate of $5 \times 10^{-5}$ with Adam, not using weight decay. We use the same tri-stage learning rate schedule, where the first 10% of iterations warm up the LR linearly from $5 \times 10^{-7}$ to $5 \times 10^{-5}$, the next 40% of iterations keep the LR constant at $5 \times 10^{-5}$, and the last 50% iterations exponentially decay the learning rate from $5 \times 10^{-5}$ to $2.5 \times 10^{-6}$. We fine-tune with a single GPU, using a batch size of 3.2M samples (200 seconds). The CNN network is frozen for all iterations, while the Transformer network is frozen for the first 5 k iterations. Masking is applied on the $\mathbf{Z}$ sequence, but only 5% of the sequence is masked. We do not apply masking on the feature dimension. We also do not use LayerDrop, as we didn't use LayerDrop during pre-training to simplify data-parallelism. Dropout is set to 10% in the Transformer layer (also during SSL).

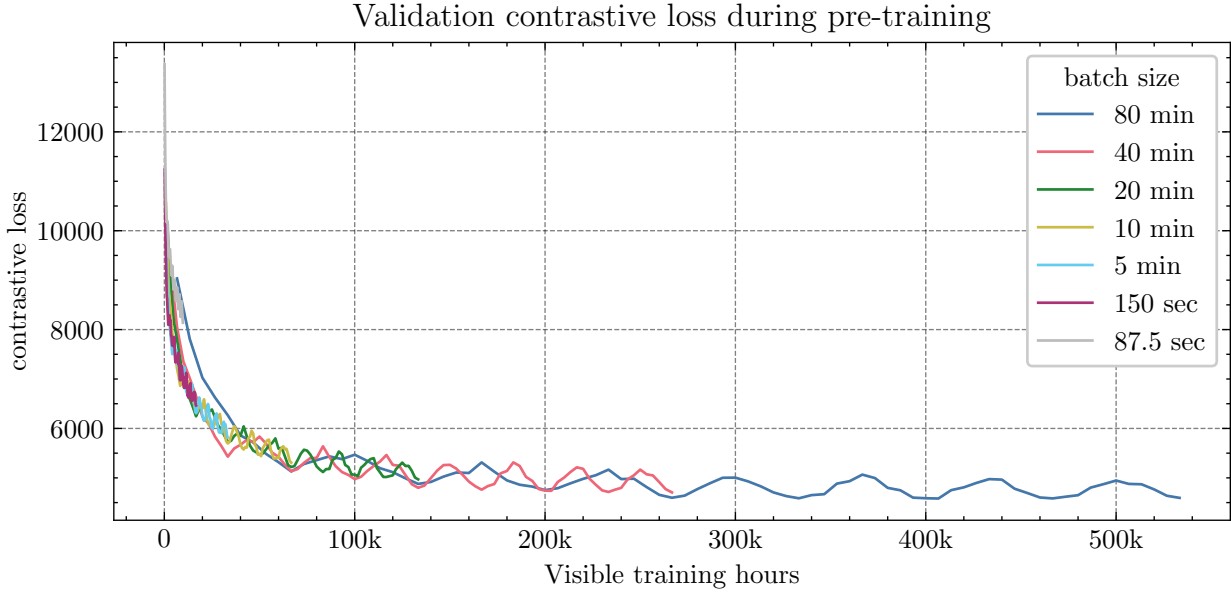

Figure 8: Validation contrastive loss

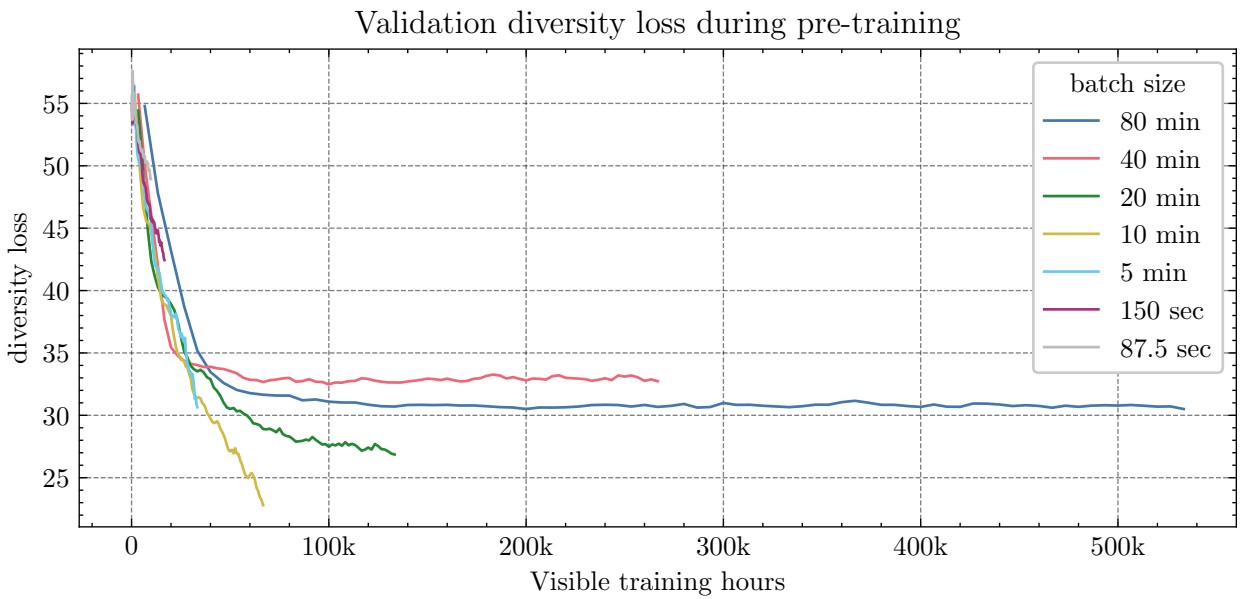

Figure 9: Validation diversity loss

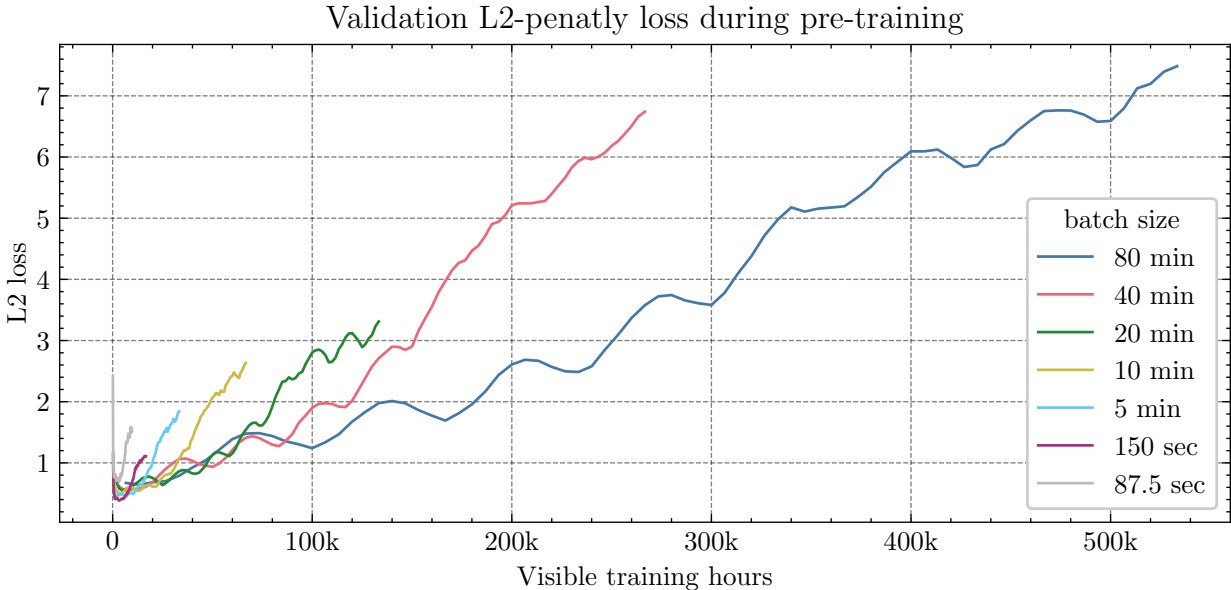

Figure 10: Validation L2-penalty loss

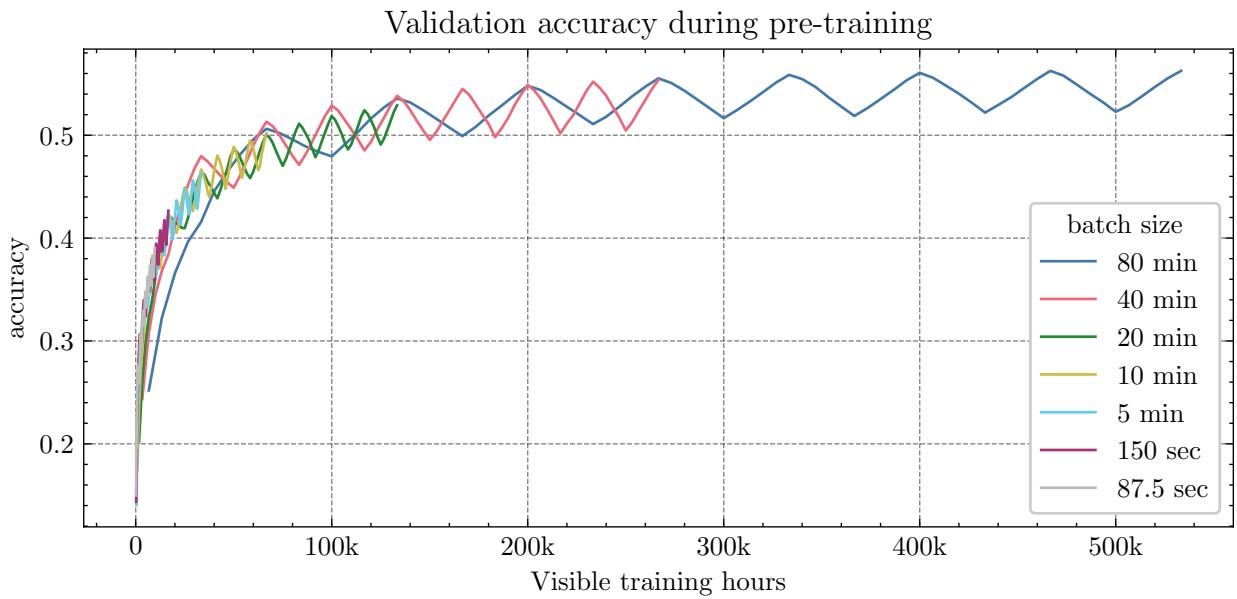

Figure 11: Validation accuracy

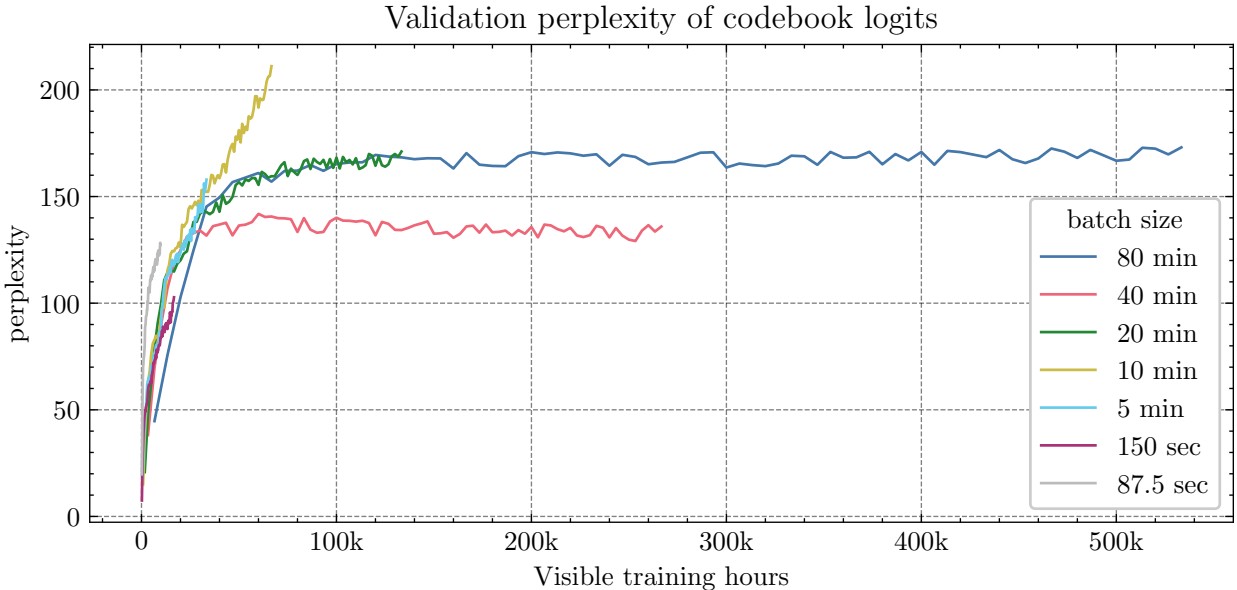

Figure 12: Validation perplexity of codebook 1

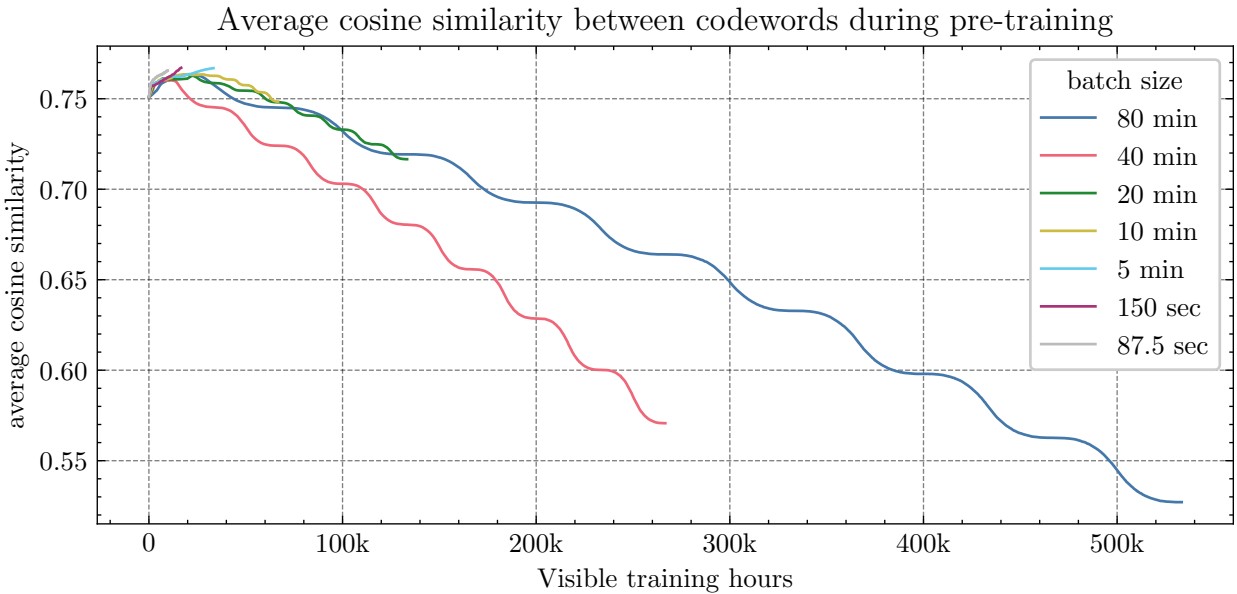

Figure 13: The average similarity between codewords in codebook 1 throughout the pre-training procedure.

## A.4 Tabular data for Figure 2

In Table 3 we show the data in Figure 2 in tabular format.

Table 3: The data seen in Figure 2. It shows WER after fine-tuning the best SSL checkpoint for each batch size. Fine-tuning is done on 5 different amounts of label conditions. Decoding is done with letter decoding as well as 4-gram word decoding with beam size 50. Scratch indicates a random initialization.

| | | letter decoding | | 4-gram word decoding | |
|---|---|---|---|---|---|
| labeled data | SSL batch size | test-clean | test-other | test-clean | test-other |
| 10 min | scratch | 129.45 | 129.47 | 111.17 | 111.95 |
| | 87.5 sec | 85.98 | 91.80 | 72.67 | 82.14 |
| | 150 sec | 72.07 | 82.02 | 54.44 | 68.26 |
| | 5 min | 61.26 | 71.06 | 40.96 | 53.53 |
| | 10 min | 50.39 | 59.45 | 32.06 | 42.32 |
| | 20 min | 47.04 | 54.67 | 27.3 | 35.96 |
| | 40 min | 45.44 | 52.05 | 26.89 | 34.35 |
| | 80 min | 41.64 | 50.06 | 24.38 | 34.47 |
| 1 hour | scratch | 106.68 | 104.76 | 101.23 | 100.09 |
| | 87.5 sec | 70.5 | 83.49 | 51.93 | 70.58 |
| | 150 sec | 53.28 | 68.37 | 34.00 | 52.00 |
| | 5 min | 39.04 | 53.57 | 22.31 | 37.11 |
| | 10 min | 28.83 | 41.47 | 15.69 | 27.03 |
| | 20 min | 25.12 | 35.23 | 13.14 | 22.25 |
| | 40 min | 23.26 | 31.75 | 11.90 | 19.52 |
| | 80 min | 21.94 | 31.30 | 11.80 | 20.72 |
| 10 hours | scratch | 105.59 | 104.03 | 100.56 | 99.25 |
| | 87.5 sec | 47.74 | 67.98 | 29.81 | 53.69 |
| | 150 sec | 34.53 | 54.43 | 20.20 | 39.96 |
| | 5 min | 23.24 | 40.24 | 13.42 | 28.07 |
| | 10 min | 16.27 | 29.51 | 9.46 | 20.27 |
| | 20 min | 13.26 | 23.83 | 7.66 | 16.31 |
| | 40 min | 11.11 | 19.78 | 6.59 | 13.46 |
| | 80 min | 10.38 | 19.56 | 6.28 | 13.67 |
| 100 hours | scratch | 69.02 | 84.70 | 50.36 | 73.44 |
| | 87.5 sec | 21.31 | 48.15 | 12.33 | 35.99 |
| | 150 sec | 16.68 | 40.41 | 9.93 | 29.63 |
| | 5 min | 12.38 | 31.43 | 7.72 | 22.66 |
| | 10 min | 9.10 | 23.43 | 5.93 | 16.87 |
| | 20 min | 7.41 | 18.93 | 5.12 | 13.72 |
| | 40 min | 6.24 | 14.87 | 4.38 | 10.88 |
| | 80 min | 5.86 | 15.97 | 4.45 | 12.05 |
| 960 hours | scratch | 25.45 | 48.65 | 12.62 | 32.20 |
| | 87.5 sec | 7.94 | 21.78 | 4.69 | 14.41 |
| | 150 sec | 7.05 | 19.60 | 4.35 | 12.77 |
| | 5 min | 6.01 | 16.84 | 4.12 | 11.45 |
| | 10 min | 5.27 | 13.91 | 3.74 | 9.66 |
| | 20 min | 4.44 | 12.53 | 3.33 | 8.90 |
| | 40 min | 4.12 | 10.62 | 3.20 | 7.87 |
| | 80 min | 4.22 | 11.58 | 3.34 | 8.68 |

## A.5 Tabular data for Figure 4

In Table 4 we show the data in Figure 4 in tabular format, where at least 2 batch sizes match exactly in the amount of data seen during self-supervision. We observe some variation between checkpoints of different batch sizes at the same amount of data seen. However, it is not evident that the larger batch size has a better WER. For example, at 16.7 k hours seen, a batch size of 150 sec has the best performance in both fine-tuning conditions, and at 50 k hours, a 10 minutes batch size outperforms 20 min batch size slightly. Except for the big performance difference at 33 k hours for the 40 minute batch size, we alleviate these differences to random noise. We hypothesize that the 40 min batch size condition performs better because we simply used a better learning rate while having minor differences in the implementation of wav2vec 2.0. We noticed after having conducted the experiments that the original wav2vec 2.0 implementation in Fairseq uses Dropout on the local speech representations before quantiziation. The original implementation also use LayerDrop on the Transformer layers, which we disabled due to it significantly slowing down the gradient synchronisation in distributed data-parallel training.

## A.6 Tabular data for Figure 5

We show the data from Figure 5 in Table 5.

Table 4: Data from Figure 4. We show the WER after fine-tuning SSL checkpoints with overlapping amount of hour seen during self-supervision, but using a different batch size and number of iterations.

| during self-supervised learning | | | fine-tuning with 10 minutes of labels (WER in %) | | fine-tuning with 100 hours of labels (WER in %) | |
|---|---|---|---|---|---|---|
| hours seen | batch size | iteration | test-clean | test-other | test-clean | test-other |
| 4.17 k | 150 sec | 100 k | 99.16 | 101.33 | 29.79 | 57.52 |
| 4.17 k | 5 min | 50 k | 98.55 | 100.04 | 30.25 | 58.14 |
| 8.33 k | 150 sec | 200 k | 88.71 | 94.32 | 22.98 | 50.31 |
| 8.33 k | 5 min | 100 k | 90.23 | 95.29 | 22.98 | 50.57 |
| 8.33 k | 10 min | 50 k | 88.35 | 93.96 | 22.35 | 49.56 |
| 12.5 k | 150 sec | 300 k | 80.83 | 88.28 | 19.23 | 44.77 |
| 12.5 k | 5 min | 150 k | 81.17 | 89.71 | 19.27 | 45.25 |
| 16.7 k | 150 sec | 400 k | 72.00 | 82.15 | 16.49 | 40.18 |
| 16.7 k | 5 min | 200 k | 73.63 | 83.54 | 16.71 | 41.17 |
| 16.7 k | 10 min | 100 k | 73.17 | 84.01 | 16.83 | 40.63 |
| 16.7 k | 20 min | 50 k | 72.95 | 83.12 | 16.89 | 41.09 |
| 25 k | 5 min | 300 k | 64.40 | 75.36 | 14.21 | 35.51 |
| 25 k | 10 min | 150 k | 63.73 | 74.55 | 13.97 | 35.50 |
| 33.3 k | 5 min | 400 k | 60.96 | 70.61 | 12.24 | 31.49 |
| 33.3 k | 10 min | 200 k | 59.00 | 69.44 | 12.46 | 31.39 |
| 33.3 k | 20 min | 100 k | 59.76 | 70.14 | 12.30 | 31.22 |
| 33.3 k | 40 min | 50 k | 53.70 | 63.52 | 10.94 | 27.53 |
| 50 k | 10 min | 300 k | 53.32 | 63.30 | 10.21 | 25.99 |
| 50 k | 20 min | 150 k | 54.04 | 63.91 | 10.47 | 26.55 |
| 66.7 k | 10 min | 400 k | 50.71 | 60.05 | 9.07 | 23.06 |
| 66.7 k | 20 min | 200 k | 51.64 | 60.64 | 9.28 | 23.97 |
| 66.7 k | 40 min | 100 k | 50.80 | 58.74 | 8.63 | 21.26 |
| 66.7 k | 80 min | 50 k | 51.26 | 60.42 | 9.29 | 23.20 |
| 100 k | 20 min | 300 k | 48.61 | 56.51 | 8.09 | 20.75 |
| 100 k | 40 min | 150 k | 48.04 | 55.41 | 7.63 | 18.67 |
| 133 k | 20 min | 400 k | 47.43 | 54.71 | 7.40 | 19.04 |
| 133 k | 40 min | 200 k | 48.06 | 55.66 | 7.11 | 17.25 |
| 133 k | 80 min | 100 k | 45.40 | 53.46 | 7.31 | 18.32 |
| 200 k | 40 min | 300 k | 46.35 | 53.42 | 6.58 | 15.74 |
| 200 k | 80 min | 150 k | 42.84 | 51.20 | 6.58 | 16.70 |
| 267 k | 40 min | 400 k | 45.57 | 52.50 | 6.26 | 15.10 |
| 267 k | 80 min | 200 k | 42.40 | 50.34 | 6.27 | 15.85 |

Table 5: SUPERB fine-tuning results shown in Figure 5.

| during self-supervision | | | PR | ASR (en) | ASR (zh) | ASV | ER | IC |
|---|---|---|---|---|---|---|---|---|
| batch size | steps | hours seen | PER | WER | CER | EER | acc | acc |
| 87.5 sec | 100 k | 2.4 k | 46.87 | 25.64 | 37.94 | 13.88 | 57.67 | 73.66 |
| 87.5 sec | 200 k | 4.8 k | 39.74 | 21.87 | 34.22 | 11.15 | 58.24 | 77.25 |
| 87.5 sec | 300 k | 7.3 k | 34.30 | 19.63 | 32.68 | 9.95 | 58.28 | 81.41 |
| 87.5 sec | 400 k | 9.7 k | 28.29 | 18.45 | 31.41 | 10.13 | 59.45 | 84.92 |
| 150 sec | 100 k | 4.2 k | 39.71 | 22.74 | 35.05 | 11.73 | 60.35 | 76.14 |
| 150 sec | 200 k | 8.3 k | 30.42 | 18.81 | 31.68 | 9.70 | 58.97 | 81.89 |
| 150 sec | 300 k | 12.5 k | 22.87 | 16.85 | 29.66 | 8.82 | 60.68 | 89.80 |
| 150 sec | 400 k | 16.7 k | 18.43 | 15.03 | 29.08 | 8.19 | 61.90 | 92.01 |
| 5 min | 100 k | 8.3 k | 30.36 | 18.75 | 31.66 | 9.33 | 59.60 | 82.44 |
| 5 min | 200 k | 16.7 k | 19.22 | 15.37 | 28.84 | 7.35 | 60.64 | 88.27 |
| 5 min | 300 k | 25 k | 14.69 | 13.62 | 27.60 | 7.30 | 61.60 | 93.75 |
| 5 min | 400 k | 33.3 k | 12.41 | 12.03 | 26.55 | 6.69 | 63.12 | 94.36 |
| 10 min | 100 k | 16.7 k | 17.57 | 14.72 | 27.82 | 7.67 | 62.11 | 91.51 |
| 10 min | 200 k | 33.3 k | 12.01 | 11.90 | 26.08 | 6.81 | 61.98 | 94.12 |
| 10 min | 300 k | 50 k | 9.71 | 10.90 | 25.29 | 6.47 | 62.69 | 95.97 |
| 10 min | 400 k | 66.7 k | 8.41 | 9.79 | 24.69 | 6.18 | 63.05 | 96.52 |
| 20 min | 100 k | 33.3 k | 12.06 | 11.85 | 26.16 | 6.80 | 63.00 | 96.20 |
| 20 min | 200 k | 66.7 k | 8.84 | 9.87 | 24.70 | 6.37 | 63.11 | 97.31 |
| 20 min | 300 k | 100 k | 7.55 | 8.84 | 24.22 | 6.15 | 63.71 | 97.36 |
| 20 min | 400 k | 133.3 k | 6.98 | 8.12 | 23.83 | 5.96 | 63.13 | 97.36 |
| 40 min | 100 k | 66.7 k | 7.94 | 9.19 | 24.76 | 5.96 | 61.88 | 96.81 |
| 40 min | 200 k | 133.3 k | 6.50 | 7.81 | 24.16 | 5.87 | 61.57 | 97.13 |
| 40 min | 300 k | 200 k | 6.37 | 7.29 | 23.97 | 5.76 | 62.57 | 96.63 |
| 40 min | 400 k | 266.7 k | 6.08 | 7.01 | 23.73 | 5.60 | 62.33 | 95.86 |
| 80 min | 100 k | 133.3 k | 6.70 | 8.02 | 23.94 | 6.08 | 62.09 | 97.50 |
| 80 min | 200 k | 266.7 k | 5.96 | 7.07 | 23.46 | 5.54 | 63.00 | 97.39 |
| 80 min | 300 k | 400 k | 5.99 | 7.11 | 23.53 | 5.58 | 62.33 | 96.41 |
| 80 min | 400 k | 533.3 k | 6.84 | 7.25 | 23.47 | 5.77 | 63.07 | 96.81 |

