# OpenReview forum: "The effect of batch size on contrastive self-supervised speech representation learning"
_TMLR — Rejected by TMLR_

### Review · Reviewer_4Evs · 2024-09-18

**Summary Of Contributions:**

The paper extensively studies the impact of varying batch sizes on the pre-training of the wav2vec 2.0 self-supervised speech representation learning model, from 87.5 seconds to 80 minutes of speech. Their findings suggest that:

1) In general larger batch sizes result in better downstream performance, if given the same amount of iteration during pre-training.
This is because they better approximate the true gradient of the objective function and also allow more data with the same number of iterations. The author(s) validated both claims by experiments.

2) The most important factor for downstream performance is the amount of data seen during SSL, rather than just the number of training iterations or the batch size in isolation, and the authors suggested benchmarking SSL algorithms in speech by constraining the amount of data seen in training.

3) In pretraining, larger batch sizes generally lead to a lower contrastive loss, a quicker dropping diversity loss, a higher L2-penalty loss, and higher accuracies.

**Audience:**

Yes

**Broader Impact Concerns:**

No, there are no major concerns over the ethical implications.

**Claims And Evidence:**

Yes

**Requested Changes:**

1) Why the critical batch size is around 40 minutes and the standard deviation barely decreases when the batch size is doubled to 80 minutes? Could the authors show some possible explanations, experiments based on the characteristics of the data, or statistical results? Further explorations may better help the community understand the effects in a nuanced way.

2) The writing could be slightly improved: for each subsection in Section 4, the authors may highlight (italic) the key takeaway from each comparison and ablations.

3) Double-check grammar or spelling, e.g., "phoneneme" in Figure 5.

**Strengths And Weaknesses:**

Strengths:
1) This paper systematically studies the effect of batch size on both pre-training and downstream performance under a reasonable computation budget and practical implications. Many downstream tasks are also considered, including phoneme recognition, ASR, OOD ASR, speaker recognition, emotion recognition, and intent classification.

2) The experiments come with many details and are carefully conducted, e.g., details of all pre-training and fine-tuning setups are stated, heuristics for learning rates are listed, standard deviation over 10 random batches averaging over all parameters computed, etc.

3) The authors seem to plan to release the code.

Weakness:

1) This paper is very practical and does not include any theoretical analyses.

---

> ### Author Response · Authors · 2024-10-22
>
> We thank the reviewer for their reading of the paper. We appreciate the comments regarding the systematic and carefully conducted experiments. We agree with the second and third requested change, our revision will include highlighting key results and we will double check spelling and grammar.
>
> Regarding the first requested change, we would like to clarify our work, which might have lead to a potential misunderstanding in the summary of the contributions. The review states that in general large batch sizes have better downstream performance because they better approximate the true gradient.  While we do observe the better gradient approximations with larger batch sizes in Figure 3, it does not necessarily follow that these lead to better performance, cf. the results from Figure 4 and 5.
> Here we see that training runs with different gradient variance have the same fine-tune performance after the same amount data seen during pre-training.  This observation is also raised by Reviewer Tioa in their 4th bullet point in WEAKNESSES.  To clarify this point, we will remove the word `critical' from ``Further, at a critical batch size of 40 minutes'' in paragraph 4.3.1.
>
> Regarding the question why the standard deviation does not decrease after a batch size of 40 minutes, we think this is interesting, but deserves a paper on its own.  We speculate, based on McCandlish et al. (2018) and Shallue et al. (2019), that the specific value is tied to the gradient noise scale, implying a correlation with the loss landscape of the objective function.  As changing training dataset or model size impacts the complexity of the loss landscape, the critical batch size should change accordingly with model size or training dataset, and for our conditions it happens to be around 40 min.  We believe that the value of the critical batch size / gradient noise scale is something that needs to be determined empirically.  We will add these considerations to Section 5.

---

> > ### Comment · Reviewer_4Evs · 2024-10-22
> > **Thank you for the response**
> >
> > The reviewer is convinced by the clear response and supports the acceptance of this paper.

---

### Review · Reviewer_Lhst · 2024-10-10

**Summary Of Contributions:**

This paper studies the effect of the batch size on model performance for contrastive self-supervised speech representation models, specifically the wav2vec 2.0 model architecture.
The authors perform a systematic comparison of the wave2vec 2.0 (`Base` version), using batch sizes ranging from 87.5 sec to 80 min. The authors compared the methods considering their training objectives, ASR fine-tuning performance, and a subset of tasks from the SUPERB challenge. The authors empirically demonstrate the importance of large batch-size for model performance across all setups.
Lastly, the authors analyzed the model w.r.t to the gradient variance during training to analyze the stability of the training / fine-tuning optimization process.

**Audience:**

Yes

**Claims And Evidence:**

No

**Requested Changes:**

As stated above, although I believe such a publication would be valuable to the speech + ML community, the scope of the current research work is limited. The authors evaluated the wav2vec 2.0 'Base' model only. It is not clear whether such findings will transfer to HuBERT and WavLM models (which are not based on contrastive loss), to CPC / wav2vec (version 1) models, or even to larger wav2vec 2.0 models. As a result, the findings presented in this paper would be valuable to researchers/practitioners working with the wav2vec `Base` version only. Additionally, as I mentioned above, the results presented in this paper are somewhat expected (larger batch size improves and stabilizes SSL models). So I expected to have a deep dive analysis of such phenomena in case there is no broader comparison.

Lastly, the writing is not fully supported by the experiments, the title of the paper is: "The effect of batch size on contrastive self-supervised speech representation learning" while only wav2vec 2.0 is considered. The intro mentions the effect of the size of the batch on SSL model performance, however only wav2vec 2.0 `Base` is considered.

To sum up, I strongly advise the authors to include additional experiments considering either more axes (different arch / different model sizes / etc.) or to perform a deeper analysis of this behavior. I know it is resource-intensive and not easy to do, however, as mentioned before, it is hard to draw general conclusions from the experimental results at this point.

**Strengths And Weaknesses:**

**Strengths:**
1. The authors performed a thorough analysis of the effect of the batch size on model performance in a fair setup. The authors considered several factor w.r.t model performance (i.e., SSL objective, ASR fine-tuning, SUPERB downstream tasks).
2. The authors additionally performed an analysis on the STD of gradients during training, considering the different batch sizes to better understand the gaps in model performance.
3. The paper is clearly written and easy to follow.

**Weaknesses:**
1. The paper considers only a single model architecture and configuration, i.e., wav2vec 2.0 'Base'.
2. The findings in this research work are pretty much expected, I'm missing some insights / additional analysis to make a more meaningful conclusion.

---

> ### Author Response · Authors · 2024-10-22
>
> We thank the reviewer for their reading of the paper. We appreciate the comments regarding the fair, thorough methodology and the writing quality.
>
> First, we would like to comment on the expectedness of the results. Our hypothesis was
> that a large batch size is required to do pre-training, as we did not find any literature on SSL pre-training with small batch sizes (e.g., using only a single, consumer-grade GPU).  Our results falsify this hypothesis, by showing that pre-training can be done with a small batch size, i.e., gradients are smooth enough to optimize the objective with a batch size that can fit on a single consumer-grade GPU. Moreover, we show that larger batch sizes have better downstream performance only because the model saw more epochs of the training data. We, at least, did not expect these results, although we had limited experience with pre-training models.
> In fact, apart from the large industrial teams that carried out the original work in SSL for speech, there has been little work published about the details of pre-training foundation models, specifically using smaller computational budgets.
> In that regard, this paper could be valuable for people with similar experience as us, or with an interest entering the field.
>
> Secondly, we understand the concern regarding the limited scope of our findings.
> We believe that
> our title, research questions and conclusions are already adequately scoped to contrastive methods, as we explicitly state that predictive SSL methods like HuBERT/WavLM/DinoSR could be analyzed in future work.  In this sense we would assign a higher priority to WavLM and HuBERT models than to the already somewhat dated wav2vec 1.0 model.
> However, to strengthen our results, we are looking into doing additional pre-trainings in the coming weeks, either with the LARGE model variant, or with a new dataset, e.g., VoxCeleb2 (~2.3k hours of training data, designed for speaker recognition).  This is likely to require an additional hyperparameter scan that may require more resources than we currently have.  As a result, we may have to be more specific in our title and conclusions that our work covers wav2vec 2.0 base only.

---

> > ### Comment · Reviewer_Lhst · 2024-10-25
> > **Official Comment by Reviewer**
> >
> > I would like to thank the authors for their detailed response.
> >
> > * I'm not sure I fully agree with the authors regarding the interpretation of the results. From Figure 2 it does seem that larger batch size reaches better performance. Results are saturated as we increase the amount of supervised training data, and I believe it is expected. For instance, when using 10min or 1 hour, it seems results are constantly improving (might reach similar performance between 40min vs 80min batch size).
> >
> > * The authors mentioned in their response: "Moreover, we show that larger batch sizes have better downstream performance only because the model saw more epochs of the training data." I'm not sure I understand where this is shown in the paper. Can the authors please clarify their comments? Do the authors mean that using a large dataset with a large batch size would perform better than a small dataset with a large batch size?
> >
> > * I highly appreciate the author's willingness to include additional experiments. I believe it will greatly improve the research work and make the results more general and of interest to the speech and ML community.

---

> > > ### Author Response · Authors · 2024-10-27
> > >
> > > Thanks for your response.
> > >
> > > > I'm not sure I fully agree with the authors regarding the interpretation of the results. From Figure 2 it does seem that larger batch size reaches better performance. Results are saturated as we increase the amount of supervised training data, and I believe it is expected. For instance, when using 10min or 1 hour, it seems results are constantly improving (might reach similar performance between 40min vs 80min batch size).
> > >
> > > The reviewer is correct in their interpretation of Figure 2. It shows that a batch size of ~40 to 80 minutes performs best across the board, independent of the amount of fine-tuning labels available, although there is a saturation effect as the amount of labels increases. Note that we fine-tune the pre-training checkpoints with the lowest validation loss, i.e., at 400k steps for all but one batch size\*. Thus, each batch size run has a different amount of seen data, that is, pre-training epochs of Librispeech, which we show in Table 2 in Appendix A1.
> > >
> > > This raises the question, do we get the same downstream task performance if we fix the amount of pre-training epochs? For example, we can pre-train for 33 Librispeech epochs by doing 100k steps with a batch size of 20 minutes or 50k steps with a batch size of 40 minutes. This analysis is done in Figure 4. We believe it convincingly shows that, independent of the batch size, the number of visible hours during pre-training is the determining factor for downstream task performance. In other words, the larger batch sizes in Figure 2 have better downstream task performance only because the models have been pre-trained for more epochs of the training data.
> > >
> > > \* For the batch size of 80 minutes the lowest validation loss was at the checkpoint of 305k steps, indicating more regularization was required for this run, or that the pretext task usefulness saturates after observing ~330k hours of the 960h Librispeech training data.
> > >
> > > > The authors mentioned in their response: "Moreover, we show that larger batch sizes have better downstream performance only because the model saw more epochs of the training data." I'm not sure I understand where this is shown in the paper. Can the authors please clarify their comments? Do the authors mean that using a large dataset with a large batch size would perform better than a small dataset with a large batch size?
> > >
> > > We hope to have clarified our quoted comment above. Regarding the dataset size, in this work we have not analyzed the size of the pre-training dataset. The current version of the paper only uses the 960h Librispeech dataset for pre-training, in accordance with the experiments using the BASE model in the original wav2vec 2.0 paper. We believe interesting future work could look into the relationship between the pre-training dataset size, and how many hours of data can be observed before the pretext task is saturated, and how this relates to downstream task performance.
> > >
> > > > I highly appreciate the author's willingness to include additional experiments. I believe it will greatly improve the research work and make the results more general and of interest to the speech and ML community.
> > >
> > > We are happy to inform the reviewer that we have reproduced the results with the LARGE model, and we will include these in the paper in the coming days.

---

### Review · Reviewer_Tioa · 2024-10-17

**Summary Of Contributions:**

This paper investigates the impact of batch size on the self-surpervised pre-training and fine-tuning of Speech models. In addition, it addresses the question of whether larger batch sizes (and associated smooth gradients) are needed to achieve the best downstream performance, or, alternatively, the amount of data observed during self-supervised learning is the dominant factor. Experiment are carried out on wav2vec2.0-base (95M parameters), with pre-training on Librispeech and fine-tuning/testing on Librispeech (speech recognition) and SUPERB benchmark (various speech tasks). Results show that larger batch sizes are beneficial to downstream performance, and that improvements are driven by the total amount of data seen by the model during pre-training, not inherent in the use of a large batch size.

**Audience:**

Yes

**Broader Impact Concerns:**

Not addressed in the manuscript. No further concern on my end.

**Claims And Evidence:**

Yes

**Requested Changes:**

See previous points.

**Strengths And Weaknesses:**

**STRENGHTS**
- Determining the best pre-training practices that influence Speech models downstream performance is an investigation of interest to the audience of this journal
- The paper is easy to follow and appropriately cites related works from the literature
- The main finding that model performance are primarily driven by the amount of data, not by the batch size, is well supported by the authors' experiments and scientifically interesting

**WEAKNESSES**
- This is an entirely empirical study, where the root causes of why batch size-dependent gradient noise has limited or no impact on downstream performance are not investigated
- The main limitation in my view is whether these findings would generalize to other model architectures and sizes. I understand pre-training is computationally expensive and this limits the range of empirical tests that can realistically be performed, but in the absence of a theoretical framework this finding might just be limited to this combination of model and tasks
- I also wonder if there are practical implications of the main finding, beyond being able to achieve similar downstream performance on a constrained hardware. Can the authors expand on this point?
- There are some mentions of a critical batch size (possibly model-dependent) beyond which performance flatten or even degrade. This would appear to break the main finding of data amount-dependent mode performance. However, the observation of this critical batch size is less supported by the data, as only seen in Fig.2 in the behavior of the models pre-trained with the highest batch size (80min). Also, this effect is no longer apparent in the data in Fig.4. Can the authors comment on this?


**OTHER COMMENTS**
- Section 3.2 states "a gpu-batch is discarded if the difference between the shortest and longest utterance is more than 10 seconds." but if utterances are ordered by length and are at most 30s long, does this ever happen? How many batches are being discarded? If this indeed happens, wouldn't this filtering affect primarily batches containing long utterances (assuming a sequence of, e.g., 1s would be grouped with utterances of similar duration) and what is the effect on pre-training quality?

---

> ### Author Response · Authors · 2024-10-22
>
> We thank the reviewer for their reading of the paper. We appreciate the comments regarding the paper's clarity and scientific validity.
>
> First, we would like to refer to our rebuttal of Reviewer Lhst, where we propose to do some additional experiments in the coming weeks in order to make our results more generalizable.
> We agree that the paper as-is only covers wav2vec 2.0 base, and we can indeed modify the title and conclusions to be more specific about this.
> However, we believe the scope of this paper is sufficient and that it is capable of sparking future work in our community.  But the reviewer is correct that for the current formulation of the conclusions, additional experiments on, e.g., speaker session heterogeneity in training data or model capacity is desirable.
>
> Secondly, to expand on the practical implications, we think that our observations will make research into (speech) SSL methods more accessible to people outside of industrial labs. This is especially true if these observations are also confirmed on other SSL methods in future work. In the least, our paper is useful as a baseline performance of pre-training on a smaller scale, e.g., new methods can be compared to wav2vec 2.0 without having to immediately pre-train on ~550 epochs of Librispeech. We hope to see that future SSL benchmarks limit the product of iterations and batch size, to make them more accessible, and to work towards more data-efficient methods.
>
> Thirdly, the review mentions a contradiction related to the critical batch size, where further increase leads to flattening or degradation of performance. We agree that our results do not support the notion that the best performance is achieved at a critical batch size. We hypothesize that the degradation observed in Figure 2 is not related to exceeding the critical batch size, but to the fact that the model starts to overfit on the training data and pretext task. This is not clearly communicated in the end of paragraph 2 of Section 5, which we will revise to include this discussion.
>
> Finally, regarding the discarding of batches, this happened only in rare edge cases, when a PyTorch DataLoader worker filling its buffer exhausted a subset with very short files and moved to one with very long files (or vice versa). We strongly believe that the frequency of this edge case does not affect the overall optimization process.

---

### Comment · Action_Editor_oaS8 · 2024-10-24
**Question regarding the optimizer scaling law**

Dear Authors,

Thanks for posting comments on the reviewers questions and concerns. I had read them and have a question regarding your comment

> Our results falsify this hypothesis, by showing that pre-training can be done with a small batch size, i.e., gradients are smooth enough to optimize the objective with a batch size that can fit on a single consumer-grade GPU. Moreover, we show that larger batch sizes have better downstream performance only because the model saw more epochs of the training data. We, at least, did not expect these results, although we had limited experience with pre-training models. In fact, apart from the large industrial teams that carried out the original work in SSL for speech, there has been little work published about the details of pre-training foundation models, specifically using smaller computational budgets.

Are you aware of the series of works on the optimizer scaling laws? It has long history on how to train with bigger batches and the same amount of epochs, see e.g. https://arxiv.org/abs/1706.02677, https://arxiv.org/abs/2205.10287. I totally agree that optimizer scaling laws were not tested / probed in speech SSL. However I think the result is not surprising if we assume that optimization is done with SGD, RMSProp, Adam for which SDE approximation is known. In such case we can go both ways: knowing how to change optimization if we want to train with bigger batches (more GPUs) for the same amount of epochs, or with small batch (say 1 GPU). We can reproduce both optimizations to follow exactly the same loss curve (if shown in the axis of number of seen data) by properly adjusting learning rate and other optimization parameters (of course only in the region where SDE approximation is correct).

Can you comment on this and what exactly new observations you bring to the table?

Could you also show the plots with the x-axis being number of visible training hours, not the iterations.

Thanks.

AE.

---

> ### Author Response · Authors · 2024-10-27
>
> Dear Action Editor,
>
> Thanks for your comments and questions. We were not aware of the work on optimizer scaling laws, and it seems appropriate to cite these works in 1) the related work section and 2) the LR heuristics explanation in Section 4.1. Thanks for pointing us in this direction. For now, we have added the requested plots, where the x-axis in Figure 1 is changed from the number of iterations to the amount of data seen, in Appendix A.2. It seems apparent that the loss values overlap for the batch sizes for which the learning rate was chosen according to square root scaling law. Notably, the lowest batch size of 87.5 seconds and the highest batch size of 80 minutes behave differently, but similar, compared to the other batch sizes. We are happy to see that we have an independent experimental confirmation that the scaling rule applies to wav2vec 2.0, and therefore also to at least one method within the speech SSL and fine-tuning paradigm.
>
> Regarding the question of what value this paper brings to the community, we believe that this paper contributes by:
>
> 1) being a reproducibility study on wav2vec 2.0, as we ablate on batch size, we share more details on the training process compared to the seminal work (e.g., training plots), we fine-tune on SUPERB, and provide a stand-alone, independent implementation.
>
> 2) sharing SSL checkpoints with a 5k iteration interval. These can be used for further analysis, e.g., studying the learning process of speech foundation models.
>
> 3) creating an extended baseline for wav2vec 2.0, so that an (initial) comparison can be made with fewer computational resources.
>
> 4) confirming scaling laws apply within the wav2vec 2.0 framework, thus showing there are no benefits to large batch sizes other than decreasing the wall-time duration of pre-training.
>
> Finally, we expect to update the paper text next week, which will include new experiments using wav2vec 2.0 LARGE, and will add changes throughout the paper to accommodate these works on scaling laws.
>
> Thanks for your time and suggestions for improvement!
>
> Kind regards,
>
> Authors

---

### Decision · Action_Editor_oaS8 · 2024-12-04

**Recommendation:** Reject

**Comment:**

Some other comments to iterate over the paper:
- SSL benchmarking with fixed amount of seen data seems incorrect to me, as this is a question how method converges. If we create the benchmark with fixed computational budget this could be more clear and practical.
- Overall batch size and iterations should be considered together and it is really about number of epochs.
- there are incorrect formulations with respect to scaling laws of optimizer in the paper, please correct them by considering seen data, epochs, not the batch size and fixed iterations.
- as I pointed hyper parameters should be changed when scale the batch (not only learning rate, weight decay also needs adjustment, and all Adam parameters as well) - right now your scaling in Table 1 is not fully correct.
- you need to dissect faster convergence (requires less epochs) and convergence for the same compute budget. For the latter we have scaling laws of optimizer and batch size change - and in this case democratization is not really democratization, as total budget the same, and we could even use gradient accumulation to reproduce wav2vec 2.0.

**Audience:**

Speech community will benefit from the paper, democratizing the SSL pretraining. Also community on scaling laws for optimizer will benefit confirming theoretical results on more empirical tasks.

**Claims And Evidence:**

The paper focuses on studying dependence between the batch size and self-supervised pretraining quality for speech, particularly focusing only on wav2vec 2.0 SSL method and having fixed number of iterations.

Ability to train with smaller batches is crucial for academy and small labs with restricted computational resources. The paper tries to reduce this gap in the community for speech as SSL is expensive so far for the reported hyper parameters.

One issue reviewers pointed out is absence of theoretical explanation of the phenomena observed by authors. However, TMLR welcomes only empirical results if they are clearly stated and supported.

The other issue is limited empirical evaluation, as authors only focused on wav2vec 2.0 model and one pretraining dataset. At the same time there is extensive downstream evaluation for the pretrained model. Authors extended revised version by having another dataset and another model size, reformulating the paper to be focused only on wav2vec 2.0.

In summary two reviewers are satisfied with the final paper state, while reviewer Lhst still thinks that observed behavior of “SSL performance on downstream tasks is heavily dependent on the amount of data the model has seen, and not necessarily the batch size.” lacks enough empirical supported experiments as this is not shown for different SSL algorithms which are now widely used, e.g. HuBERT.

In general, I think it will be ok to have the paper focused only on wav2vec 2.0 to report first observations as this could be of significant help to community to have other pretrained checkpoints of wav2vec 2.0 and democratizing it across different labs and academia. However, I have a huge issue with the way empirical analysis is conducted as authors were unaware of scaling laws for optimizers, and moreover failed to incorporate this known result into the paper: it is included as related work and some search of learning rate and graphs in appendix, but authors did not do in depth analysis, proper hyper parameter tuning, correct usage of scaling law for adamw, and proper derivations of conclusions. In general, we already know scaling laws for optimizers, which means that with the same compute budget = number of epochs we can train with any batch till the same performance. We also know that SSL loss in general correlated with the downstream performance, though after some training this doesn’t hold necessary. What authors show is that we do not depend on batch size (known from scaling law of optimizers) but depend on the number of seen data = epochs (known as if SSL loss is lower more likely downstream performance is better).

Having this, I believe authors should revise the paper and possible versions could be:
- consider analysis of scaling laws of optimizers for SSL - then the paper should be revisited in methodology, proper plots having x-axis as number of epochs, adamw scales not only learning rate, but also b1, b2, eps parameters which authors failed to do. In the end I expect to see all plots showing the similar loss curves for SSL losses and for downstream performances if the proper parameters of optimizer are set when scaling the batch size. Don’t forget that learning rate schedule should be also properly adjusted, e.g. warmup should be measured in epochs not iterations to have proper scaling.
- change narrative and content and report contributions like reproduction, providing checkpoints, some new observations.

**Resubmission Of Major Revision:**

The authors may consider submitting a major revision at a later time.